# Gene therapy via canalostomy approach preserves auditory and vestibular functions in a mouse model of Jervell and Lange-Nielsen syndrome type 2

Xuewen Wu [1,2], Li Zhang[2,3], Yihui Li[4], Wenjuan Zhang[3], Jianjun Wang[2], Cuiyun Cai [1,2] & Xi Lin[2✉]

Mutations in voltage-gated potassium channel *KCNE1* cause Jervell and Lange-Nielsen syndrome type 2 (JLNS2), resulting in congenital deafness and vestibular dysfunction. We conducted gene therapy by injecting viral vectors using the canalostomy approach in *Kcne1*[−/−] mice to treat both the hearing and vestibular symptoms. Results showed early treatment prevented collapse of the Reissner's membrane and vestibular wall, retained the normal size of the semicircular canals, and prevented the degeneration of inner ear cells. In a dose-dependent manner, the treatment preserved auditory (16 out of 20 mice) and vestibular (20/20) functions in mice treated with the high-dosage for at least five months. In the low-dosage group, a subgroup of mice (13/20) showed improvements only in the vestibular functions. Results supported that highly efficient transduction is one of the key factors for achieving the efficacy and maintaining the long-term therapeutic effect. Secondary outcomes of treatment included improved birth and litter survival rates. Our results demonstrated that gene therapy via the canalostomy approach, which has been considered to be one of the more feasible delivery methods for human inner ear gene therapy, preserved auditory and vestibular functions in a dose-dependent manner in a mouse model of JLNS2.

[1] Department of Otolaryngology Head and Neck Surgery, Xiangya Hospital of Central South University, 87 Xiangya Road, 410008 Changsha, Hunan, China. [2] Department of Otolaryngology, Emory University School of Medicine, 615 Michael Street, Atlanta, GA 30322, USA. [3] Department of Otolaryngology, Union Hospital, Tongji Medical College, Huazhong University of Science and Technology, Wuhan, China. [4] Department of Pharmacy, Changsha Hospital of Traditional Medicine, 22 Xingsha Avenue, 410100 Changsha, Hunan, China. ✉email: xlin2@emory.edu

It is estimated that about 5% of the world's population (approximately 432 million adults and 34 million children) suffers from disabling hearing loss defined as >40 dB hearing loss in at least one of the ears. Sensorineural hearing loss (SNHL) accounts for 50–60% of the affected patients, and genetic mutations in deafness genes are responsible for about half of SNHL cases. Mutations in >100 human genes have been identified to cause non-syndromic and syndromic hearing loss (https://hereditaryhearingloss.org/). Genetic disorders with only the vestibular symptoms are rare, as most genetic peripheral vestibular disorders are also associated with hearing loss[1], and examples include DFNA9, 11, 15, and 28[2], as well as DFNB16[3]. Both auditory and vestibular functions may be affected in patients with either Usher syndrome[4], enlarged vestibular aqueduct syndrome[5], or Jervell and Lange-Nielsen syndrome (JLNS)[6,7]. About 35% of adults aged >40 years have some degree of balance dysfunction in the United States according to the National Health and Nutrition Examination Survey[8]. Disorders of balance functions substantially worsen quality of life and increase the burden on healthcare systems. While attention has been mostly given to the treatment of hearing loss in recent inner ear gene therapy studies[9–11], relatively few studies have focused investigation on the efficacy of gene therapy for treating both cochlear and vestibular disorders[4,12].

The voltage-gated KCNQ1/KCNE1 potassium channel complex plays an essential role in transepithelial voltage gradient and potassium ion movement in several organs, including the heart, kidney, colon, small intestine, and the inner ear[13]. In the inner ear, KCNQ1/KCNE1 potassium channels are expressed exclusively in the apical membrane of marginal cells (MCs) in the stria vascularis (SV) and in vestibular dark cells[14,15]. Mutations in either KCNQ1 or KCNE1 subunit cause the JLNS, and inactivating either Kcnq1 or Kcne1 in mice produces a model of human JLNS1 or JLNS2, respectively[7,16,17]. Both $Kcne1^{-/-}$ and $Kcnq1^{-/-}$ mice exhibit an abnormal development of the endolymphatic space, severe degeneration of the hair cells (HCs), and spiral ganglion neurons (SGNs)[18,19]. These mutant mice show severe hearing and balance dysfunctions[19,20]. Here we used $Kcne1^{-/-}$ mice to investigate whether a modified adeno-associated virus subtype 1 (named AAV1-CB7-Kcne1) expressing the wild-type (WT) Kcne1 in the inner ear could preserve the morphology of the inner ear and functions of both the auditory and vestibular systems in these mutant mice when the AAV1-CB7-Kcne1 was injected into the posterior semicircular canal (PSCC) by the canalostomy approach. The secondary outcomes in mice (e.g., growth rate, birth rate, and survival rate) were also monitored. Our results showed that injection of AAV1-CB7-Kcne1 into the PSCC of neonatal (postnatal day 0–2 [P0-P2]) $Kcne1^{-/-}$ mice, in a dose-dependent manner, preserved the normal morphology of the inner ear. The hearing and vestibular functions in the mutant mice were also improved for at least 5 months in the high-dosage treatment group. In the mice treated with low dosage, most of them showed improvement in vestibular function in the absence of hearing preservation.

## Results

### Injection of viral solution between P0 and P2 via the canalostomy approach results in extensive on-target and ectopic expression without damaging hearing and vestibular functions in WT mice. Virally mediated green fluorescent protein (GFP) expression levels in the inner ear were first used to examine viral transduction efficiency. AAV1-CB7-GFP was injected into the PSCC of WT mice (P0–P2, Supplementary Fig. 1 and Supplementary Video 1). Examinations performed at P30 (n = 6) showed normal gross cochlear and vestibular morphologies. GFP expression was found in >95% inner hair cells (IHCs) (Fig. 1a,

d–f). In contrast, outer hair cells (OHCs) were only sparsely GFP-positive (Fig. 1d–f). GFP-labeled cells were also found in 77.5 ± 2.9% of MCs in the SV (Fig. 1c), which were the targeted cells for treatment in this study. GFP-positive cells were also found in SGN regions (Fig. 1a, b) and vestibular compartments (Fig. 1g–l). Extensive GFP signals were confirmed to be in the SGNs (Fig. 1m, n) by specific labeling with antibody against NF200 and in vestibular HCs that were identified by specific labeling with antibody against Myosin 7a (Myo7a) (Fig. 1o–r). GFP-positive cells were also found in apparent supporting cells in the vestibular compartments, including the saccule (Fig. 1g, h, o, p), utricle (Fig. 1i, j, q, r), and crista ampullaris (CA) (Fig. 1k, l). To determine whether our surgical procedures and virally mediated ectopic gene expression impaired normal hearing and vestibular functions, we measured auditory brainstem responses (ABRs) and vestibular functions of injected WT mice at P30. Results showed that ABR thresholds (Supplementary Fig. 2a, b), peak amplitudes, and latencies of the ABR waves were not changed comparing to uninjected control ears (n = 6, p > 0.05 for all tested parameters, Student's t test; Supplementary Fig. 2c, d). There were also no differences in vestibular behaviors after PSCC injections, including circling behavior, rotarod performance, and swimming ability (n = 6 in each group, p > 0.05 for all tested parameters, Student's t test; Supplementary Fig. 2e–h). These results support the conclusion that our surgical procedure for virus delivery through injections into the PSCC and the extensive viral-mediated GFP expression did not impair hearing and vestibular functions in WT mice.

Kcne1 is normally expressed on the apical membrane of MCs and vestibular dark cells (Fig. 2a)[14,15]. Consistent with the expected results from the $Kcne1^{-/-}$ mice, no Kcne1 expression was detected in the MCs and vestibular dark cells of untreated $Kcne1^{-/-}$ mice (Fig. 2b). After injecting 1.5–2.0 μL (high dosage) of AAV1-CB7-Kcne1 into the PSCC of $Kcne1^{-/-}$ mice, immunolabeling showed on-target expression in the MCs (75.7 ± 4.4%, n = 4) and in the vestibular dark cells (36.0 ± 6.1%, n = 4) (Fig. 2d). Mice received low-dosage (0.5–1.0 μL) injections of AAV1-CB7-Kcne1 showed Kcne1 expression levels in notably fewer MCs (29.9 ± 5.3%, n = 4) and vestibular dark cells (15.7 ± 2.8%, n = 4) (p < 0.05 for all tested parameters, Student's t test; Fig. 2c). Extensive ectopic expression of Kcne1 were also found in the vestibular end organs (e.g., utricle and CA) in both high- and low-dosage groups of $Kcne1^{-/-}$ mice (Supplementary Fig. 3). These results demonstrate that viral inoculation into the PSCC transduces both targeted and untargeted cells in a dose-dependent manner. In addition, ABR thresholds were unchanged in WT mice injected with AAV1-CB7-Kcne1 (Supplementary Fig. 2b), which supports the conclusion that ectopic Kcne1 expression does not affect normal hearing. These results were also consistent with our previous findings with the ectopic expression of Kcnq1[10].

When AAV1-CB7-GFP was injected later at P30 into the WT mice (n = 6), many inner ear cells (e.g., vestibular dark cells and IHCs) were still transduced at the adult stage (Supplementary Fig. 4a–c), although the MCs in the SV were not transduced at P30 suggesting adult treatment may not be successful. The ABRs, measured 7 days after injections, were normal in the injected adult WT mice (n = 6; Supplementary Fig. 4d), supporting that the injection procedures done at the adult stage did not damage hearing. Apparently due to the abnormal development of the semicircular canals in $Kcne1^{-/-}$ mice that give rise to much smaller canals and degenerated vestibular membrane at the adult stage (Fig. 3), we could not successfully inject viral solution in the adult $Kcne1^{-/-}$ mice. In addition, injections of high dosage of AAV1-CB7-Kcne1 at P3 in $Kcne1^{-/-}$ mice (n = 4) did not yield comparable hearing improvements as those injections at P0–P2

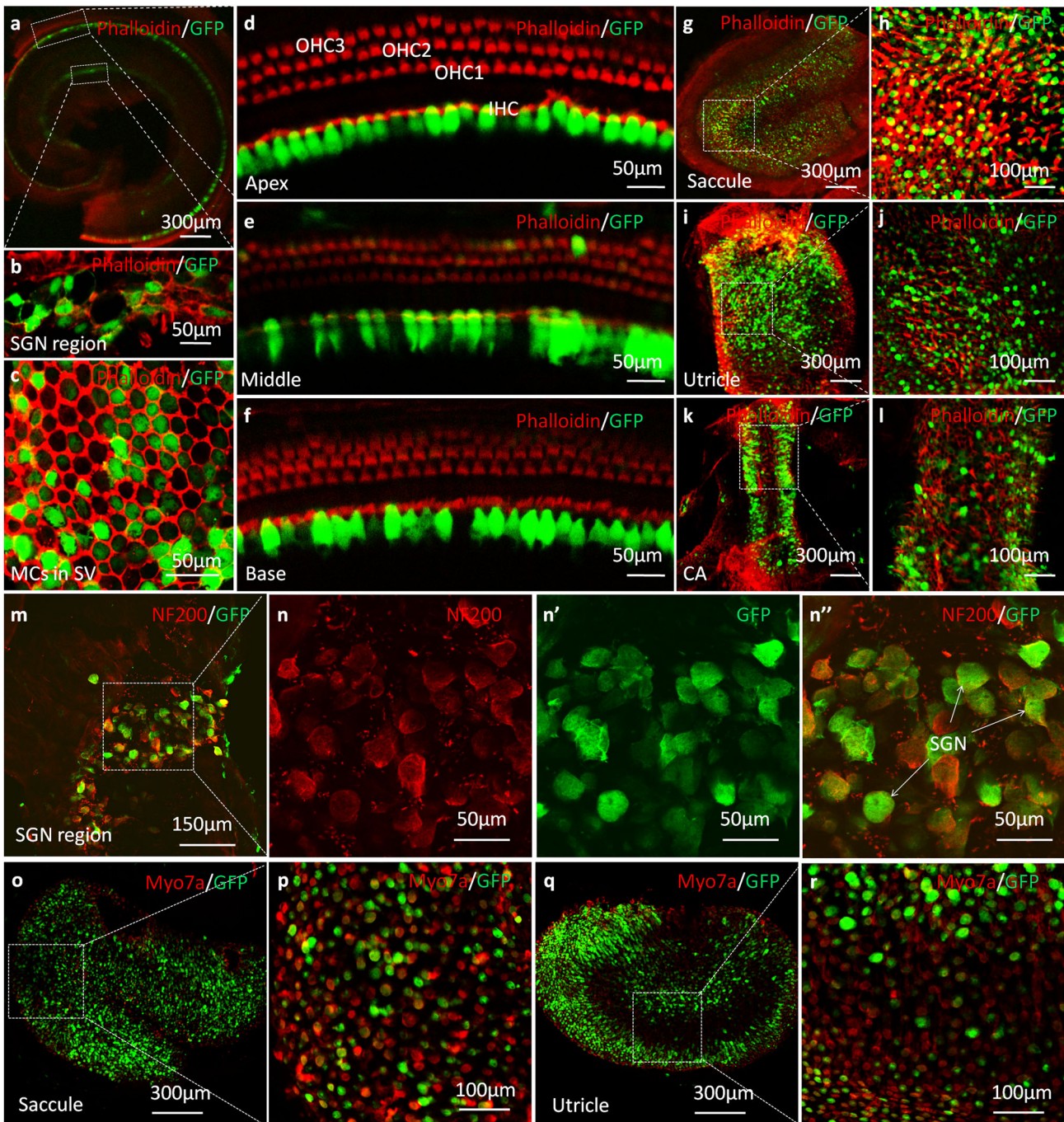

**Fig. 1 Confocal immunofluorescent images of whole-mount preparations of cochleae and vestibules in WT mice at 1 month after AAV1-CB7-*GFP* delivery via PSCC injections at P0–P2 (green: GFP; *n* = 6 in each group). a** A representative low-magnification image of an apical middle turn. **b** A high-magnification image of the spiral ganglion neuron (SGN) tissue. **c** A high-magnification image of marginal cells (MCs) in the stria vascularis (SV). **d–f** High-magnification images of the apex, middle, and base of the basilar membrane. **g–j** Representative low- and high-magnification images of saccule and utricle. **k**, **l** Representative low- and high-magnification images of the crista ampullaris (CA). **m** Representative low-magnification images of SGNs co-labeled with antibodies against NF200 (red) and GFP. **n**, **n″** Representative high-magnification images of SGNs (boxed area in **m**) labeled with antibodies against NF200 (**n**, red) and GFP (**n′**), **n″** is the superimposed images of **n** and **n′**. **o–r** Representative low- (**o**, **q**) and high (**p**, **r**) magnification images of saccule and utricle co-labeled with antibodies against Myo7a (red) and GFP.

(Supplementary Fig. 5). We therefore focused on our studies about gene therapy of the *Kcne1*$^{-/-}$ mice to those injected between P0 and P2.

**Virally mediated *Kcne1* expression prevents severe morphological defects in both the cochlear and vestibular compartments of the *Kcne1*$^{-/-}$ mice.** *Kcne1*$^{-/-}$ mice were injected with AAV1-CB7-*Kcne1* into the left PSCC at P0–P2 and the inner ear morphologies of both ears were examined at P30. In untreated ears (Fig. 4b), most cochlear and vestibular HCs were severely damaged as indicated by loss of ciliary bundles. The cell border and cell size of MCs were often irregular (Fig. 4b). These were in sharp contrast to the regular hexagonal shape of the normal MCs in the SV (Fig. 4a). In *Kcne1*$^{-/-}$ mice treated with the low dosage

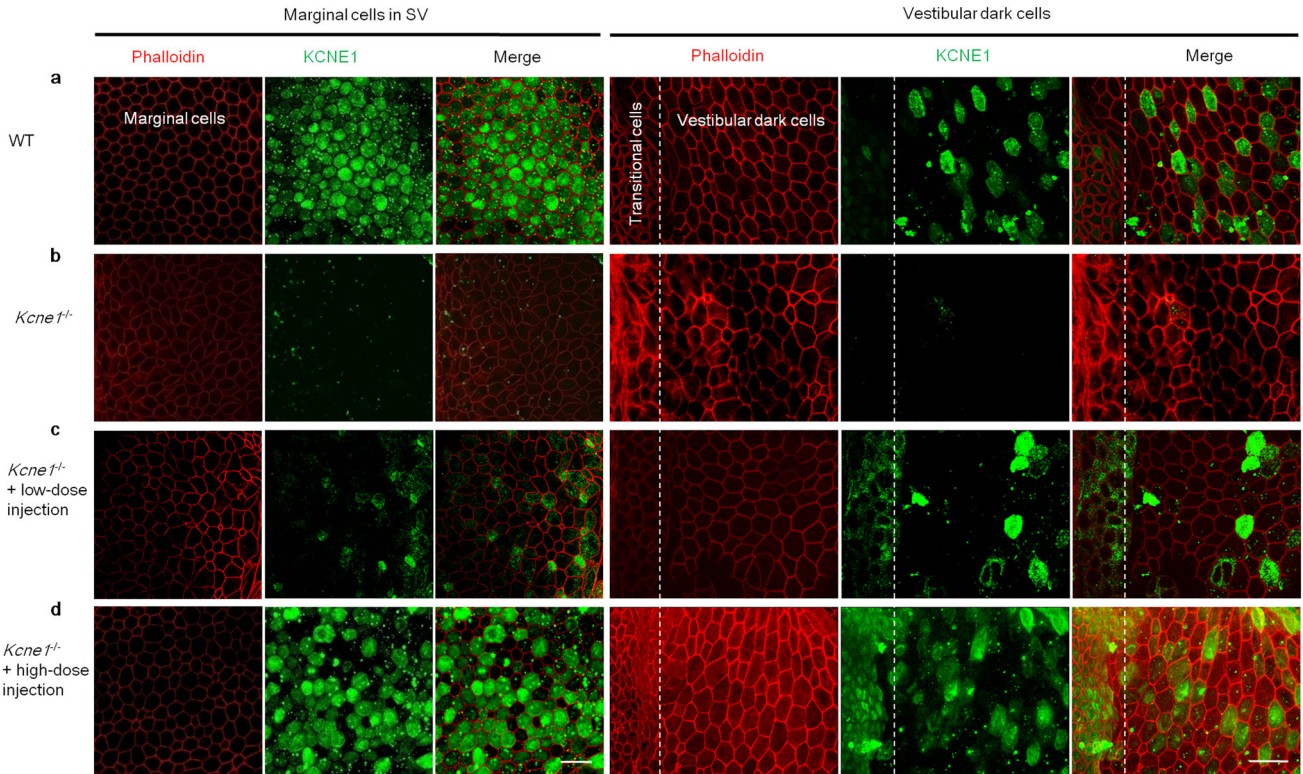

**Fig. 2 Comparison of *Kcne1* expression levels in marginal cells (MCs) and vestibular dark cells in WT mice, untreated *Kcne1*$^{-/-}$ mice, and low- and high-dose-treated *Kcne1*$^{-/-}$ mice at P30 ($n = 4$ in each group). a** *Kcne1* expression in MCs and vestibular dark cells in WT mice. **b** Absence of *Kcne1* expression levels in MCs and vestibular dark cells in untreated *Kcne1*$^{-/-}$ mice. **c** *Kcne1* expression levels were found in some MCs and vestibular dark cells in low-dose-treated *Kcne1*$^{-/-}$ mice. **d** Extensive *Kcne1* expression levels in MCs and vestibular dark cells in high-dose-treated *Kcne1*$^{-/-}$ mice. Scale bar is 50 μm and applies to all panels. Details of data quantifications are given in the text. Red: phalloidin; Green: KCNE1.

(Fig. 4c), the normal morphologies of ciliary bundles of cochlear IHCs and vestibular HCs in the utricle and CA appeared to be preserved, although nearly all OHCs and vestibular HCs in the saccule, as well as the MCs in the SV, were apparently damaged (Fig. 4c). These were in sharp contrast to the results we obtained from mice injected with high-dosage viral solution. The ciliary bundles of cochlear HCs and vestibular HCs, as well as the hexagonal shape of the MCs in the SV, appeared to be normal in the high-dose-treated ears of *Kcne1*$^{-/-}$ mice (Fig. 4d).

The bodies of cochlear HCs were labeled with an antibody against Myo7a to further observe the morphological changes of cochlear HCs after AAV1-CB7-*Kcne1* injection. In WT cochleae, the shape and arrangement of IHCs and OHCs displayed their normal patterns (Supplementary Fig. 6a–c). However, both types of cochlear HCs were severely degenerated in all turns in the untreated ears of *Kcne1*$^{-/-}$ mice (Supplementary Fig. 6d–f). In the low-dose-treated ears of *Kcne1*$^{-/-}$ mice (Supplementary Fig. 6g–i), the number and shape of IHCs were apparently normal in all turns. However, most OHCs appeared to be degenerated in the middle and basal turns. After injections with the high dosage of AAV1-CB7-*Kcne1*, IHCs and OHCs in all cochlear turns appeared to be normal (Supplementary Fig. 6j–l). These results were consistent with the morphological changes we observed with labeling of ciliary bundles of cochlear HCs by phalloidin (Fig. 4).

One of the major morphological defects in the inner ears of *Kcne1*$^{-/-}$ mice was that the diameter of the semicircular canals was significantly smaller (Fig. 3a). We observed that our treatment with the high-dosage viral solution was able to correct this phenotype. The average outer diameter of the bony superior semicircular canal (SSCC) in the treated ears of *Kcne1*$^{-/-}$ mice was almost twice as large (256.6 ± 8.4 μm) as that of the untreated

ears (114.1 ± 8.0 μm) ($n = 8$ in each group, $p < 0.0001$, Student's $t$ test; Fig. 3b, d), which brought the average outer diameter of the SSCC in the treated ears of *Kcne1*$^{-/-}$ mice similar to that of WT mice (269.6 ± 7.0 μm, $n = 8$ in each group, $p = 0.65$, Student's $t$ test; Fig. 3d). Consistent with these results, gross morphology of the vestibular membranous labyrinth after the treatment (Fig. 3c) appeared to be larger, although it was problematic to accurately quantify changes in the size of the soft vestibular membranous labyrinth. The treatments also corrected the reduction in the cavities of the three semicircular canals. The cross-sectional areas of these cavities were much smaller in the untreated ears of *Kcne1*$^{-/-}$ mice (Supplementary Fig. 7a, b), which were 0.013 ± 0.003, 0.018 ± 0.003, and 0.038 ± 0.005 mm$^2$ for lateral semicircular canals (LSCC), SSCC, and PSCC, respectively ($n = 4$; Supplementary Fig. 7d). In WT mice (Supplementary Fig. 7a, b), these values were 0.038 ± 0.009, 0.045 ± 0.006, and 0.110 ± 0.011 mm$^2$ for LSCC, SSCC, and PSCC, respectively ($n = 4$ in each group, comparisons between the two groups were $p = 0.03$ for LSCC, $p = 0.007$ for SSCC, and $p = 0.0009$ for PSCC, Student's $t$ tests; Supplementary Fig. 7d). We found that the cross-sectional areas of the cavities of the LSCC, SSCC, and PSCC after treatment in *Kcne1*$^{-/-}$ mice with the high dosage were similar to those of WT mice ($n = 4$ in each group, $p > 0.05$ for all comparisons, Student's $t$ tests; Supplementary Fig. 7a, b). The specific values in ears of treated *Kcne1*$^{-/-}$ mice were 0.030 ± 0.007, 0.038 ± 0.003, and 0.103 ± 0.011 mm$^2$ for LSCC, SSCC, and PSCC, respectively ($n = 4$ in each group). All these values were about 2–3-fold larger than those of the ears of untreated *Kcne1*$^{-/-}$ mice ($p = 0.058$ for LSCC, $p = 0.001$ for SSCC, and $p = 0.002$ for PSCC, Student's $t$ tests). In addition, *Kcne1*$^{-/-}$ mice exhibited collapsed vestibular wall surrounding the CA

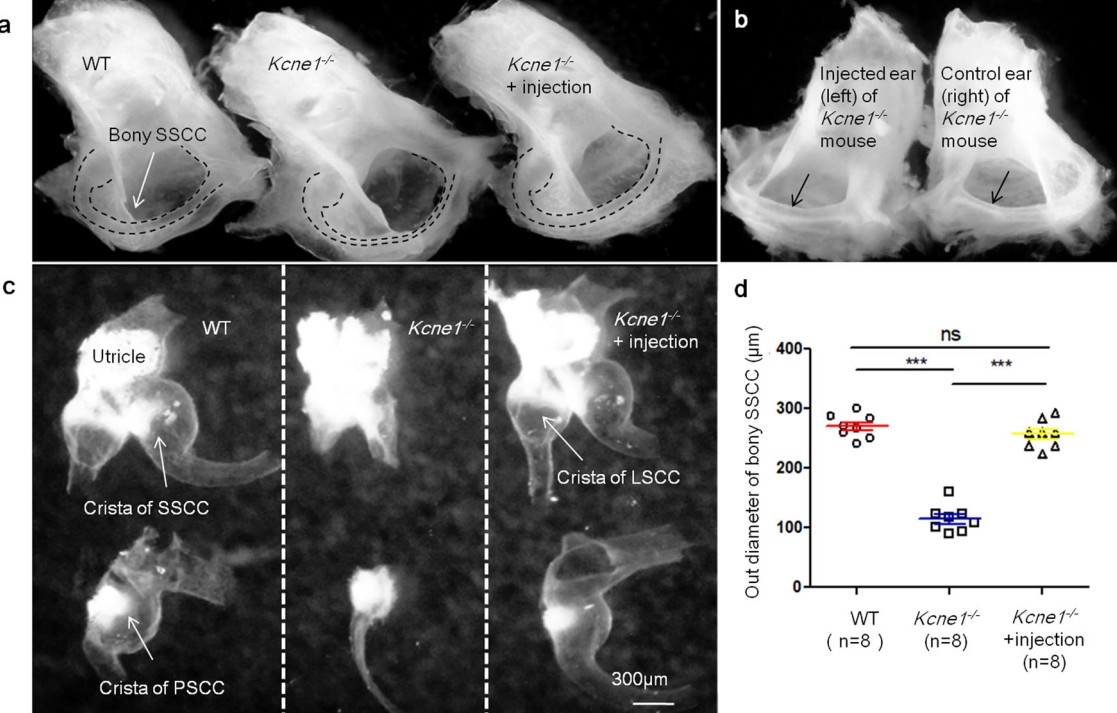

**Fig. 3 Morphology of semicircular canals and vestibular membranous labyrinths in WT, untreated Kcne1$^{-/-}$ mice, and treated Kcne1$^{-/-}$ mice at P30 (n = 8 in each group). a** Comparison of the gross morphologies of bony semicircle canals in WT, untreated Kcne1$^{-/-}$ mice, and treated Kcne1$^{-/-}$ mice. Border of bony superior semicircular canal (SSCC) is outlined by dashed lines. **b** Gross morphologies of bony posterior semicircular canal (PSCC) in left and right ear of treated and untreated ears of the same Kcne1$^{-/-}$ mouse. Arrows point the region where the diameters of the semicircular canals were measured. **c** Gross morphologies of vestibular membranous labyrinths in WT, untreated Kcne1$^{-/-}$ ear, and treated ears of Kcne1$^{-/-}$ mice. **d** Quantification of the outer diameter of bony SSCC in WT, untreated Kcne1$^{-/-}$ mice, and treated Kcne1$^{-/-}$ ears as labeled (n = 8 in each group, p < 0.0001 in WT ears, and p < 0.0001 in treated Kcne1$^{-/-}$ ears, comparing to untreated Kcne1$^{-/-}$ ears, two-sided Student's t tests). ***: p < 0.001. Data are shown as mean ± SEM. Source data are provided as a Source data file.

(Supplementary Fig. 7c), which was prevented after treatment. The location of the vestibular wall was similar to that found in the WT mice (Supplementary Fig. 7c).

Compared to that in the WT cochlea (Fig. 5a–d), the Reissner's membrane was collapsed in untreated Kcne1$^{-/-}$ mice, which resulted in a much smaller scala media (SM) space (Fig. 5a). The thickness of the SV was much thinner in untreated Kcne1$^{-/-}$ mice (16.5 ± 1.6 μm, n = 4) than that in age-matched adult WT mice (30.5 ± 1.0 μm, n = 4) (p = 0.0003, Student's t test; Fig. 5a–c). The cochlear HCs and SGNs were extensively damaged in untreated Kcne1$^{-/-}$ mice (Fig. 5a–d). Treatment with the high dosage of AAV1-CB7-Kcne1 dramatically prevented damages to various types of inner ear cells (examined at P30). The normal position of the Reissner's membrane, the thickness of the SV (29.8 ± 1.1 μm, n = 4) (p = 0.0004, compared to the untreated group, Student's t test), and the normal stereocilia of HCs in the organ of Corti and the general appearance of SGNs were all preserved (Fig. 5d, e). These findings showed that the treatments with the high dosage were effective in correcting defective morphological phenotypes found in both the cochlea and vestibular compartments of Kcne1$^{-/-}$ mice.

To monitor treatment efficacy over a longer time period, we next examined the morphologies of inner ears at 6 months (P6m) after injections (n = 6; Supplementary Fig. 8). Almost all cochlear and vestibular HCs, as well as the MCs, were severely damaged in untreated Kcne1$^{-/-}$ mice when examined at P6m (Supplementary Fig. 8a). In the low-dosage group (Supplementary Fig. 8b), only a few vestibular HCs showed normal ciliary bundles in the utricle and CA at P6m. Severe damage or degeneration was also found in cochlear HCs, MCs, and vestibular HCs in the saccule. In the high-dosage group at P6m, the morphologies of the inner

ears in two mice with partial hearing preservation were still normal (Supplementary Fig. 8c top row). We also observed that cellular degeneration of cochlear HCs, MCs, and vestibular HCs occurred at different levels in mice with partial hearing preservation or with poor hearing preservation (Supplementary Fig. 8c, middle and bottom rows, respectively). These results suggested a correlation between the hearing and morphological preservations and support that the treatment effects were retained for 6 months in some of the treated Kcne1$^{-/-}$ mice, although the reason for variation in hearing preservation is unclear.

**Virally mediated gene therapy by injection into the PSCC prevents hearing loss in Kcne1$^{-/-}$ mice.** Kcne1$^{-/-}$ mice did not show any detectable ABRs at sound levels up to 100 dB sound pressure level (SPL), which was the loudest sound that could be generated by our equipment (Fig. 6). In Kcne1$^{-/-}$ mice treated with low dosage, we observed a few dB improvement in hearing thresholds. Compared to those of the untreated ear, ABR thresholds (tested 30 days after injections) improved by 2.0 ± 0.2, 3.5 ± 0.4, 3.5 ± 0.3, and 2.0 ± 0.2 dB SPL at 8, 12, 18, and 24 kHz (n = 20; Fig. 6b), respectively. In the high-dosage group, we observed more noticeable improvements in hearing thresholds. The differences in ABR thresholds between the injected and un-injected ears in the same animal averaged 15.0 ± 3.2, 37.0 ± 3.4, 44.0 ± 3.6, 42.0 ± 3.4, 22.0 ± 2.8, and 8.5 ± 1.7 dB SPL at 4, 8, 12, 18, 24, and 32 kHz, respectively (n = 20; Fig. 6b). The average peak I amplitude of the ABRs in the high-dosage group (n = 20) tested at 90 dB SPL were 65.0, 65.8, 87.2, 73.6, 63.7, and 29.4% of those of the WT mice (n = 8) at 4, 8, 12, 18, 24, and 32 kHz,

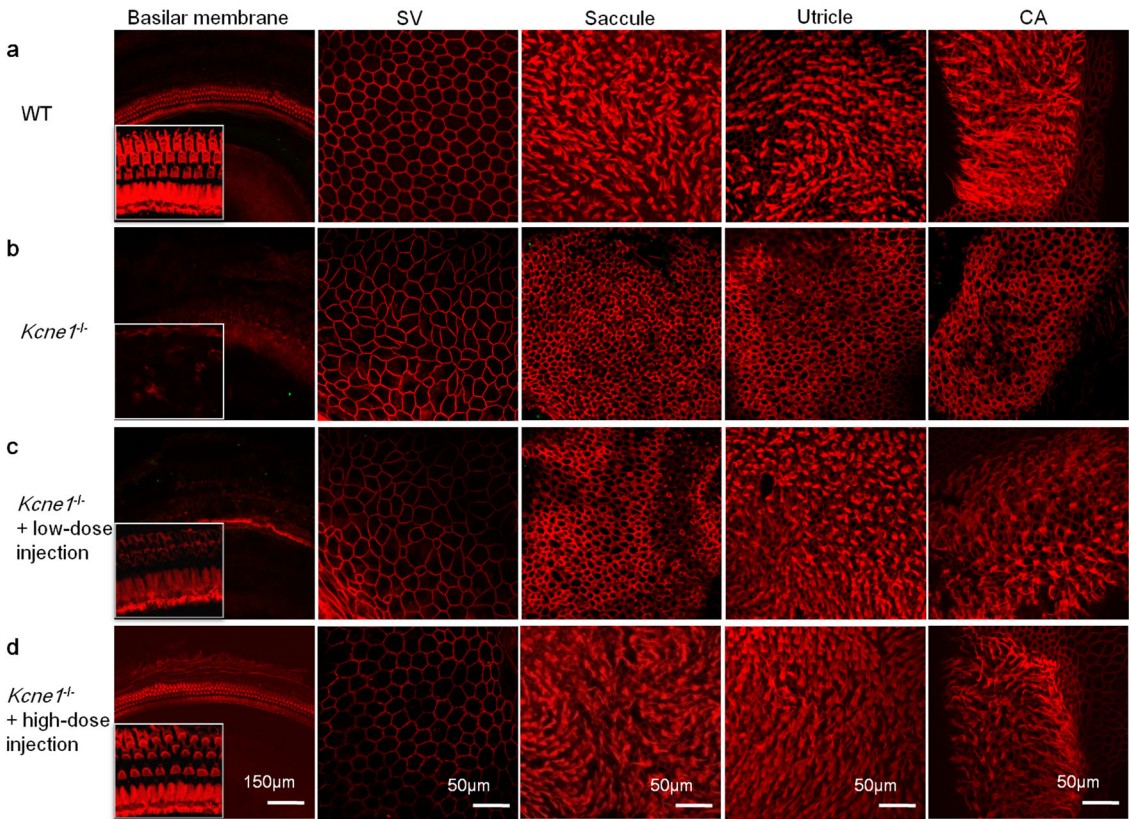

**Fig. 4 Isothiocyanate-conjugated phalloidin labels both cell membrane and ciliary bundles for comparisons of the morphologies of the WT, untreated, and low- and high-dose-treated *Kcne1*$^{-/-}$ mice at P30 (*n* = 4 in each group). a** The morphologies of marginal cells (MCs), ciliary bundles of cochlear, and vestibular hair cells (HCs) in WT mice as labeled. **b** The morphologies of MCs, ciliary bundles of cochlear, and vestibular HCs in untreated *Kcne1*$^{-/-}$ mice. **c** The morphologies of MCs, ciliary bundles of cochlear, and vestibular HCs in low-dosage-treated *Kcne1*$^{-/-}$ mice. The ciliary bundles of vestibular HCs in the utricle and crista ampullaris (CA) were significantly preserved. **d** The morphologies of MCs, ciliary bundles of cochlear, and vestibular HCs in high-dosage-treated *Kcne1*$^{-/-}$ mice. The shape of the MCs and the ciliary bundles of the HCs in the cochlear and vestibular organs seemed to have recovered to normal. Red: phalloidin.

respectively (Fig. 6c). Long-term monitoring of ABRs in the high-dosage group revealed that ABR thresholds of all frequencies tested at 5 months after injections were little changed comparing to those tested at 1 month (*n* = 6, *p* > 0.05, Student's *t* test; Fig. 6d). Although the ABR improvement tested at 6 months after injections started to show declines, apparent hearing improvements remained in some frequencies. The average threshold differences between treated and untreated ears tested at 6 months were 18.3 ± 7.9, 28.3 ± 10.1, 28.3 ± 10.1, and 15.0 ± 6.2 SPL at 8, 12, 18, and 24 kHz, respectively (*p* < 0.05 for all frequencies; Fig. 6d). These results support that the treatment efficacy on hearing preservation in the high-dosage treatment group of mice by our delivery method was noticeable and stable for up to 5 months, and the treatment efficacy started to decline thereafter.

**Gene therapy by the PSCC injection approach preserves vestibular functions in *Kcne1*$^{-/-}$ mice.** *Kcne1*$^{-/-}$ mice exhibited frequent head tossing, gait instability, and circling behavior, as well as occasional head tilting (Supplementary Video 2). We observed qualitatively that the treated *Kcne1*$^{-/-}$ mice in both the low- and high-dosage groups exhibited improvements in head and gait stability, as well as in normal explorative behavior without head tilting (Fig. 7a). The high-dosage group seemed to show a longer-lasting efficacy. These observations supported that our treatment preserved vestibular functions (Supplementary Videos 3–4). We quantified the frequencies of circling by

calculating the number of rotations that each mouse made in 2 min segments (tested at P30), which were 0 ± 0 times in WT mice (*n* = 8), 93 ± 16 times in untreated *Kcne1*$^{-/-}$ mice (*n* = 12), 4.5 ± 4.3 times in low-dose-treated *Kcne1*$^{-/-}$ mice (*n* = 6), and 0 ± 0 times in high-dose-treated *Kcne1*$^{-/-}$ mice (*n* = 6, comparison between untreated and treated *Kcne1*$^{-/-}$ mice, *p* < 0.0001, Student's *t* test; Fig. 7 and Supplementary Videos 3–4). After 6 months, the average number of rotations in the high-dosage group of treated *Kcne1*$^{-/-}$ mice was still less than that observed in the low-dosage treatment group of mice (*n* = 6 in each group, *p* = 0.04, Student's *t* test; Fig. 7a). Vestibular functions were further tested using a rotarod apparatus, which measures how long mice can hold onto a rotating rod without falling off. WT mice (*n* = 8) stayed on the rotating rod for 120 ± 0 s (the longest time tested) and 100.8 ± 10.5 s at rotating speeds of 5 and 20 rpm, respectively. In comparison, untreated *Kcne1*$^{-/-}$ mice (*n* = 12) fell after 8.9 ± 1.8 and 2.5 ± 0.9 s at rotating speeds of 5 and 20 rpm, respectively. Upon testing at 1 month after treatments in the high-dosage group (*n* = 6; Fig. 7b, c), mice maintained balance on the rotarod for 111.4 ± 8.6 and 79.8 ± 14.8 s at rotating speeds of 5 and 20 rpm, respectively. Moreover, the improvement in the high-dose-treated mice lasted for at least 6 months (Fig. 7b, c). In contrast, low-dose-treated mice showed a gradual decline in rotarod performance for both the low- and high-speed tests (*n* = 6; Fig. 7b, c).

We also scored the swimming ability of mice as another independent test of vestibular function according to a method

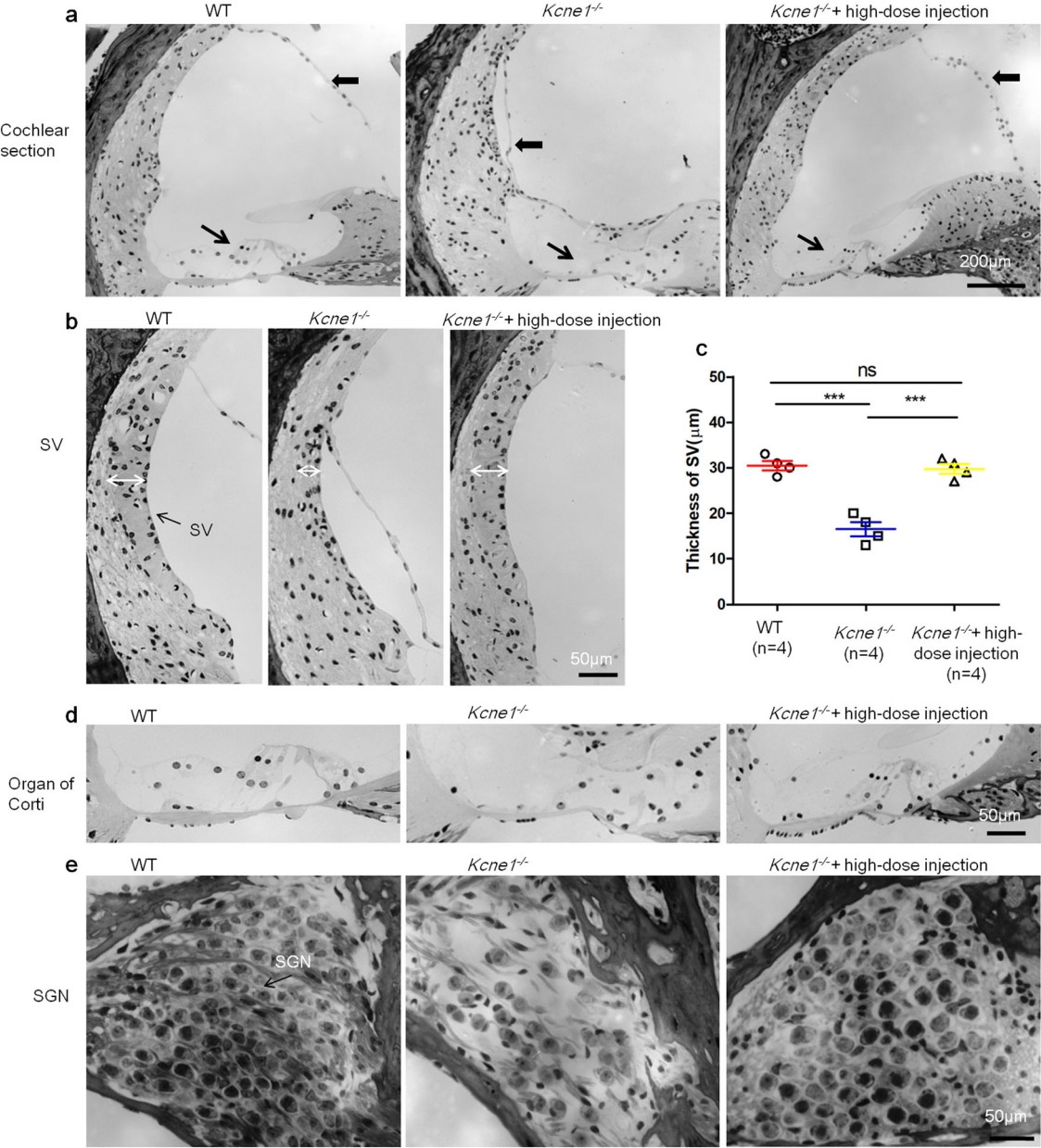

**Fig. 5 Morphologies of cochlear sections in the middle turn of WT, untreated, and high-dose-treated $Kcne1^{-/-}$ mice observed at P30 ($n = 4$ in each group). a** Comparison of the morphology of the cochlear section of WT, untreated, and high-dose-treated $Kcne1^{-/-}$ mice. **b** Comparison of the thickness of the stria vascularis (SV) in the middle turn of the WT, untreated, and high-dose-treated $Kcne1^{-/-}$ mice. **c** Quantitative comparison of the thicknesses of the SV in the WT, untreated, and high-dose treated $Kcne1^{-/-}$ mice ($n = 4$ in each group, $p = 0.0003$ in WT ears and $p = 0.0004$ in treated $Kcne1^{-/-}$ ears, comparing to untreated $Kcne1^{-/-}$ ears, two-sided Student's $t$ tests). Data are shown as mean ± SEM. ***: $p < 0.001$. Source data are provided as a Source data file. **d** Comparison of the morphology of the organ of Corti of the WT, untreated, and high-dose-treated $Kcne1^{-/-}$ mice. **e** Comparison of the morphology of spiral ganglion neurons (SGNs) in the middle turn of the WT, untreated, and high-dose-treated $Kcne1^{-/-}$ mice. Heavy black arrows point to the Reissner's membrane. Fine black arrows point to the organ of Corti. White double-headed arrows indicate the border for measuring the thickness of the SV.

introduced by Isgrig et al.[4]. All WT mice had scores of 0 ($n = 8$; Fig. 7d and Supplementary Video 5). Untreated $Kcne1^{-/-}$ mice had swimming scores of 3 ($n = 12$; Fig. 7d and Supplementary Video 6). In comparison, treated $Kcne1^{-/-}$ mice had swimming scores of 1.7 ± 0.4 ($n = 6$) and 0.5 ± 0.3 ($n = 6$) in the low- and high-dosage treatment groups of $Kcne1^{-/-}$ mice tested at P30, respectively ($p < 0.001$ for all comparisons, Student's $t$ tests; Fig. 7d and Supplementary Video 7). The results showed that the treatment improved vestibular functions in $Kcne1^{-/-}$ mice in a dose-dependent manner (Fig. 7d). Although the swimming scores of treated mice worsened gradually over a period of 6 months,

these scores in the high-dose-treated group were still better than those of untreated $Kcne1^{-/-}$ mice at 6 months ($p = 0.003$, Student's $t$ test; Fig. 7d). With PSCC injection approach, we observed that 65% (13/20) of treated mice showed vestibular improvements without hearing improvements in the low-dosage group. In the high-dosage group, 100% (20/20) of treated mice showed vestibular improvements and 80% (16/20) of treated mice showed hearing improvements. None of the mice showed hearing preservation without displaying improvement in the vestibular function. These results support that vestibular dysfunction in $Kcne1^{-/-}$ mice was improved after injecting the AAV1-CB7-

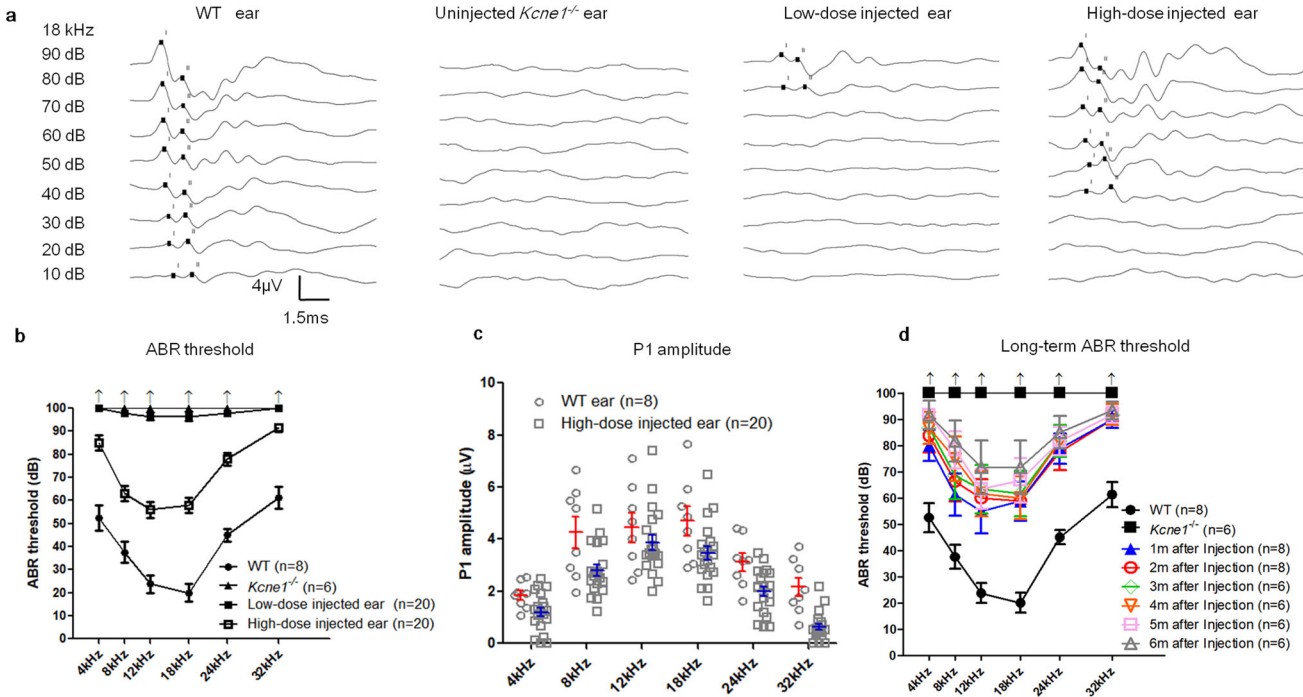

**Fig. 6 Comparison of auditory brainstem response (ABR) data measured from WT, untreated, and treated Kcne1$^{-/-}$ mice ($n \geq 6$ in each group).**
**a** Series of representative waveforms of ABRs elicited by different intensities of tone burst (at 18 kHz) are given for WT, untreated, and low- and high-dose-treated Kcne1$^{-/-}$ mice (as labeled). **b** Averaged ABR thresholds at various frequencies for different groups of mice as indicated by the figure legends. Data are shown as mean ± SEM. Upward arrows indicate that ABR thresholds were at the maximal sound level that could be measured by our equipment. Source data are provided as a Source data file. **c** The peak amplitude of ABR wave 1 (P1) elicited at 90 dB SPL from 4 to 32 kHz. Source data are provided as a Source data file. **d** Changes in ABR thresholds of high-dose-injected Kcne1$^{-/-}$ ears measured at 1–6 months after injections. Data are shown as mean ± SEM. Source data are provided as a Source data file.

Kcne1 viral construct into the PSCC, even for mice in the low-dosage group that displayed no hearing improvement.

**Secondary outcomes improved by viral-mediated gene therapy through the PSCC injections.** We hypothesized that constant circling behavior and other vestibular dysfunctions may affect the growth, breeding productivity, and offspring survival rates among Kcne1$^{-/-}$ mice, and gene therapy may alleviate these phenotypes. Compared to that of WT mice, the untreated Kcne1$^{-/-}$ mice nursed by Kcne1$^{-/-}$ mothers had a lower body weight at 4 (14.2 ± 0.4 g) and 6(18.3 ± 0.4 g) weeks of age ($n = 6$, $p = 0.007$ at 4 weeks, $p = 0.02$ at 6 weeks, Student's $t$ tests). In contrast, the untreated Kcne1$^{-/-}$ mice nursed by WT mothers (although their hearing and vestibular functions were not improved) and the treated Kcne1$^{-/-}$ mice in both the low- and high-dosage groups nursed by WT mothers showed similar weight growth comparing to WT mice during 8-week period ($n = 6$, $p > 0.05$ in all comparisons, Student's $t$ tests; Supplementary Fig. 9a). During a 6-month mating period, 4 WT breeding pairs produced 5–6 litters each, or ~1 litter/month. When breeding productivity was compared, we found that 4 WT breeding pairs produced a total of 22 litters during the 6-month period, whereas 4 untreated Kcne1$^{-/-}$ breeding pairs produced a total of 11 litters during the same period. In comparison, 4 high-dose-treated Kcne1$^{-/-}$ breeding pairs and 4 low-dose-treated Kcne1$^{-/-}$ breeding pairs produced 17 and 16 litters during the 6-month period, respectively ($p = 0.035$ in the high-dosage group and $p = 0.047$ in the low-dosage group, comparing to the untreated group, Student's $t$ tests; Supplementary Fig. 9b). In addition, the survival rate among litters born to the treated Kcne1$^{-/-}$ breeding pairs was increased. On average, 90.8 ± 5.3% of litters from WT parents survived to their weaning age of P21, but only 25 ± 14.4% of mice born to untreated Kcne1$^{-/-}$ parents survived to the same

weaning age ($n = 4$ pairs in each group, $p = 0.005$, Student's $t$ tests; Supplementary Fig. 9c). In comparison, 60.8 ± 9.8% of litters born to low-dose-treated Kcne1$^{-/-}$ parents and 73.8 ± 9.4% of litters born to high-dose-treated Kcne1$^{-/-}$ parents survived until the weaning age ($n = 4$, $p = 0.043$ in the low-dosage group and $p = 0.017$ in the high-dosage group, comparing to untreated group, Student's $t$ tests; Supplementary Fig. 9c). These results showed that both high-dosage and low-dosage treatment had better offspring birth rates and survival rates comparing to the untreated Kcne1$^{-/-}$ mice. The reason for these improvements is unclear, although we speculate that better vestibular functions may play a role; however, much work remains to be done to clarify the underlying mechanism.

## Discussion
JLNS is an autosomal-recessive hereditary condition with major symptoms of congenital bilateral SNHL and long Q-T in electrocardiographic patterns[21]. In clinic, some JLNS patients also display vestibular dysfunction[22]. Currently, no effective therapeutics are available for treating JLNS. We carried out gene therapy by delivering a viral vector (AAV1-CB7-Kcne1, with high and low dosages) through injections into the PSCC of Kcne1$^{-/-}$ mice, which is the first preclinical trial of inner ear gene therapy for the JLNS2. Null mutations in either Kcne1 or Kcnq1 disrupt endolymph production and K$^+$ circulation in the inner ear, causing deafness and vestibular dysfunction[19,20]. Vetter et al. reported collapse of the Reissner's membrane and degeneration of the organ of Corti across all cochlear turns of neonatal Kcne1$^{-/-}$ mice as early as P3[19]. Consistent with these findings[19], we also found the collapse of the Reissner's membrane and extensive degeneration of multiple types of inner ear cells (e.g., cochlear and vestibular HCs, MCs, and SGNs) in Kcne1$^{-/-}$ mice. Following major factors were considered

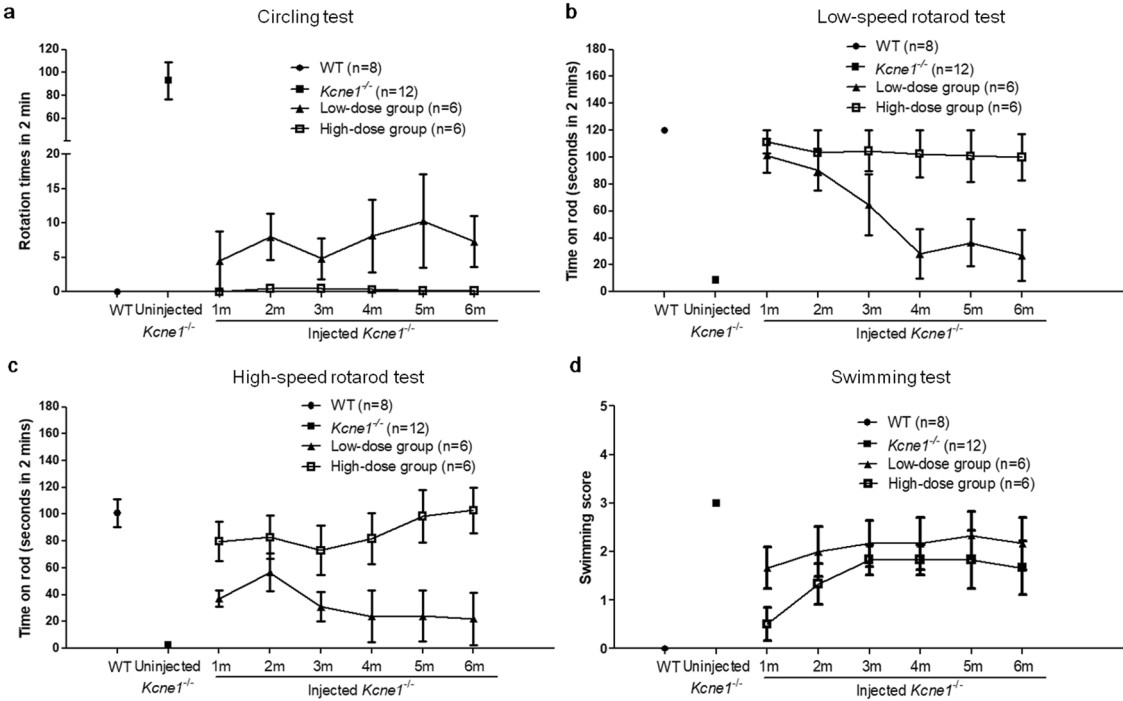

**Fig. 7 Examinations on the vestibular functions of the WT, untreated, and low- and high-dose-treated *Kcne1*$^{-/-}$ mice measured at 1–6 months after injections ($n \geq 6$ in each group). a** Changes in the number of rotations of in WT, untreated, and low- and high-dose-injected *Kcne1*$^{-/-}$ mice as indicated by the figure legends. Data are shown as mean ± SEM. Source data are provided as a Source data file. **b** Changes in rotarod performance (low-speed tests) of WT, untreated, and low- and high-dose-injected *Kcne1*$^{-/-}$ mice. Data are shown as mean ± SEM. Source data are provided as a Source data file. **c** Changes in rotarod performance (high-speed tests) of WT, untreated, and low- and high-dose-injected *Kcne1*$^{-/-}$ mice. Data are shown as mean ± SEM. Source data are provided as a Source data file. **d** Changes in swimming scores of WT, untreated, and low- and high-dose-injected *Kcne1*$^{-/-}$ mice. Data are shown as mean ± SEM. Source data are provided as a Source data file.

in our experimental designs and data analyses: (1) gene therapy implemented after degeneration of cochlear cells would likely fail[23]; (2) our injections done at P3 failed to give effective treatment results in hearing preservation (Supplementary Fig. 5); and (3) the extremely small diameter of the PSCC in untreated adult *Kcne1*$^{-/-}$ mice made the injections done to the adult PSCC unsuccessful (Fig. 3). We therefore focused analyzing treatment results to those performed in a time window between P0 and P2.

Our studies demonstrated apparent preservation of both auditory and vestibular functions by delivering viral particles through the canalostomy approach, which is a gene delivery route considered to be a likely candidate for human studies[7,17]. Our data suggest that, after inoculation by canalostomy approach through the PSCC, the viral construct can diffuse into the SM of the cochlea, transfect target cells (e.g., MCs in the SV) in early postnatal period, and also induce extensive ectopic expression in the inner ear. The on-target expression of *Kcne1* was detected in 75.7% of MCs and 36% of vestibular dark cells 30 days after injection of AAV1-CB7-*Kcne1*. The results suggested that treatment performed between P0 and P2 alleviated cellular damage or degeneration in both the cochlear and vestibular components. Our finding that functional preservation by injections into the PSCC was only effective in early postnatal stages (P0–P2) is consistent with previous studies showing that only early postnatal interventions were able to yield positive results for hearing preservation in multiple mouse models of congenital hearing loss (e.g., *Vglut3*$^{-/-}$ mice[9], *Kcnq1*$^{-/-}$ mice[10], *Ush1c c.216G>A* mice[24], and *Otof*$^{-/-}$ mice[11]). The results suggest that 100% viral transduction of targeted cells was not a requirement for achieving significant treatment efficacy. In the high-dosage group, normal morphologies of the Reissner's membrane, the organ of Corti, MCs, SGNs, and cells in the vestibular end organs were preserved.

Furthermore, treated mice showed apparent improvements in hearing thresholds across frequencies from 8 to 24 kHz. In a subgroup of treated mice, the improvement of the hearing threshold was as high as 60 dB SPL at 12 kHz. Apparent morphological and very limited hearing preservation were observed in mice that received low-dosage injections (Figs. 4c and 6b). Other important findings of this study were that the degree of the preservation of inner ear morphology and auditory function depended on the dosage of viral vectors, and the therapeutic effect on hearing in the high-dosage group was stable for 5 months. We showed that ectopic expression of *Kcne1* in the inner ear did not affect hearing threshold of WT mice tested at 1 month (Supplementary Fig. 2a, b), although future studies are required to determine whether ectopic expression of *Kcne1* in the inner ear (e.g., in HCs) can gradually lead to hearing loss.

When vestibular functions were evaluated, we found that both low-dose-treated (even in mice without hearing improvements, 13 out of 20 mice) and high-dose-treated *Kcne1*$^{-/-}$ mice exhibited fewer circling behaviors, as well as better performances on both rotarod and swimming tests 30 days after injections. These improvements were dose dependent, with high-dose-treated *Kcne1*$^{-/-}$ mice showing more stable efficacies. The results demonstrate that efficacies on hearing and vestibular phenotypes may display independent outcomes after treatments. The PSCC injection route may preferentially yield vestibular therapeutic efficacy, which suggest that severe and disabling vestibular dysfunctions caused by genetic mutations may be treated by gene therapy via PSCC injections when no hearing preservation is expected.

There are three major injection routes for performing inner ear gene therapy, including injections through the SM, round window membrane (RWM), and via canalostomy (e.g., PSCC, LSCC)[23] approaches. Many studies of inner ear gene delivery in

mouse models utilize the trans-cochlear pathway (e.g., SM, or RWM) to deliver exogenous genes[9,12,24]. SM injection achieves extensive expression of exogenous genes in the cochlear cells lining the SM. However, this injection route requires a delicate surgical protocol at early developmental stages, and such surgical procedure is likely to cause severe hearing loss in the more mature and ossified cochlea[25]. Compared to the delivery route through the SM, viral inoculation into the scala tympani through the RWM is atraumatic in neonatal mice but shows less efficient gene expression in cochlear and vestibular cells in the sensory regions, such as those cells lining the endolymphatic space. An ideal inoculation method for inner ear gene therapy to treat cochleovestibular disorders should be both atraumatic and permit widespread transduction of multiple types of targeted cells specifically throughout the cochlea and vestibular end organs. The PSCC or LSCC is accessible for relatively easy surgical approaches. The canalostomy is considered to be one of the more feasible delivery methods for future human inner ear gene therapy without causing significant functional damage[4,26,27]. Recent studies have utilized the trans-vestibular pathway (e.g., canalostomy) for cochleovestibular gene delivery, most of which have investigated GFP expression rather than the expression of therapeutic genes[26–29], except for two recent studies in which one used a model of Usher syndrome (Whirlin[−/−] mice)[4] and the other one used Cx30 knockout mice[30]. Important similarities and differences exist among these two previous studies and our present study, and these are summarized in Table 1. One important difference is that the cellular targets for treatment were different, MCs and vestibular dark cells in the present study, HCs in Isgrig et al. study, and non-sensory cells and fibrocytes in Crispino et al. study. Another important outcome difference is that, in addition to the restoration of normal vestibular function, treated mice exhibited apparent hearing improvements in our study (e.g., 40–70 dB SPL at 8–18 kHz in high-dose-treated Kcne1[−/−] mice). Compared to

other studies, a 0–10 dB SPL improvement at 32 kHz was found in treated Cx30[Δ/Δ] mice[30], a 0–40 dB SPL improvement at 8 kHz was found in treated Whirlin[−/−] mice[4], and no hearing improvement was found in treated Cx30[−/−] mice[30]. Furthermore, the treatment efficacies reported in our present study were more substantial in terms of hearing threshold preservation[4,30]. These differences in the degrees of hearing improvements in different models (Kcne1[−/−], Cx30[Δ/Δ], Cx30[−/−], and Whirlin[−/−]) are likely due to differences in genes targeted and the time courses of morphological and hearing deteriorations associated with each gene. One common finding among published studies is in the transient nature of treatment efficacies. Even in the high-dosage group of mice in our study, we observed that hearing improvements started to decline at 5 months after treatment. In contrast, both morphologies of inner ears (Supplementary Fig. 8c) and the ABR thresholds (Fig. 6d) in two mice that received the high-dosage injections were still preserved at P6m. More cellular degeneration appeared in the cochlear HCs, MCs, and vestibular HCs in mice that showed partial hearing preservation. The degree of cellular degeneration appeared to be correlated with severity of hearing loss (Supplementary Fig. 8c). Possible explanations for the reason of efficacy decline may include a gradual decline of the virally mediated Kcne1 expression or excessive ectopic expression of Kcne1 with toxic effect over time in cells that Kcne1 is not expressed endogenously, such as HCs. One way to test this hypothesis is to use the Kcne1 promoter to drive the expression specifically in the inner ear. In Isgrig et al.[4], this decline occurred at <4 months after treatment. We observed that Kcne1[−/−] mice did not breed well, and the birth rate and the survival rate of offspring were significantly reduced when they did breed. A novel finding of our studies is that AAV1-mediated Kcne1 gene therapy improved secondary outcomes in treated Kcne1[−/−] mice.

Considering that AAV1-CB7-GFP could not transduce the adult MCs of mice in the present study, it seems to be necessary to

**Table 1 Comparison of gene therapy results obtained by the canalostomy injection method in the current study and two published studies.**

|  | This study | Crispino et al. | Isgrig et al. |
|---|---|---|---|
| Viral subtype | AAV1 | BAAV | AAV8 |
| Target gene | Kcne1 | Gjb6 | Whirlin |
| Promoter used | CBA | CMV | CMV |
| Virus injection time | P0–P2 | P4 | P4 |
| Injection route | PSCC | LSCC | PSCC |
| Repeated with different batches of viral solution | Yes | Unclear | Yes |
| Dose | Low-dosage group: 0.5–1.0 µL; high-dosage group: 1.5–2.0 µL | 1.0 µL | 0.98 µL |
| Targeted cells | Marginal cells and vestibular dark cells | Non-sensory cells of the sensory epithelium and fibrocytes in the supra-strial zone | Cochlear and vestibular hair cells |
| Ectopic expression of therapeutic gene | Yes and extensive | No | No |
| Expression of GFP | Yes | Yes | Yes |
| Hearing improvement | 0–30 dB SPL at 8–18 kHz in the low-dosage group; 40–70 dB SPL at 8–18 kHz in the high-dosage group | Slight improvement (0–10 dB SPL) at 32 kHz in Cx30[Δ/Δ] mice; no improvement in Cx30[−/−] mice | 0–40 dB SPL at 8 kHz |
| Vestibular function assessment | Apparent improvement in vestibular function as measured by circling, rotarod, and swimming test | Unclear | Apparent improvement in vestibular function as measured by circling, rotarod, swimming, and vestibular-evoked potential test |
| Long-term treatment effect | At least for 5 months | Unclear | Less than 4 months |
| Intracellular trafficking | At the apical membrane of marginal cells and vestibular dark cells | In the cell membrane of supporting cells | At the top of cilia bundles of hair cells |
| Secondary outcomes | Improvement in birth and litter survival rates | Unclear | Unclear |

identify new serotypes of AAVs or other means to transduce adult MCs in future human translational studies. Our results suggested that gene therapy implemented after degeneration of cochlear and vestibular cells is unlikely to be successful if the cells in the inner ear are already degenerated no matter which injection route is utilized. Due to the very limited availability of materials, the pathological changes about the morphologies in the cochleae of JLNS2 patients still remain unknown. We are not sure whether therapeutic efficacy can be achieved by gene therapy in the cochlea of newborn babies of JLNS2 patients. However, the postnatal window of efficacy for the gene therapy in the mouse model suggests that the window to achieve therapeutic efficacy in humans may be in the second trimester (around 18 weeks gestational age in humans) and prior to hearing onset in humans[31]. This implication may generate additional social and ethical issues for the human treatment given the nonlethal nature of the disease.

In summary, our results support that inner ear gene therapy using the canalostomy approach effectively preserved vestibular and hearing functions in mouse models, and the treatment efficacy for the vestibular phenotypes may be independently displayed. However, genetic hearing loss with balance dysfunction in humans is uncommon (e.g., due to compensation by the visual system). The underlying causes for a majority of clinical balance problems are unclear. We thus cannot make a conclusion that the current gene therapy approach demonstrated in the $Kcne1^{-/-}$ mouse model via canalostomy approach is widely applicable to treat patients with balance problems. Given the extent and duration of the functional recovery displayed here using morphological, physiological, and behavioral assessments, our results suggest that $Kcne1$ gene replacement therapy for recessive $KCNE1$ mutations in human JLNS2 patients is promising for further development of cochlear and vestibular gene therapies. To advance toward clinical translational goals, future studies are also needed to investigate longer-term efficacy and immunoreactivity of AAVs in inner ear gene therapy.

## Methods

**Preparation of viral constructs for injections**. pAAV1-CB7-$Kcne1$ and pAAV1-CB7-GFP plasmids (maps shown in Supplementary Fig. 1) were verified by restriction digestion. Details of the production method of AAV constructs have been described in one of our published papers[10]. Briefly, recombinant AAV1 particles were produced by double transfection of HEK293T cells with the AAV and AAV-helper packaging plasmids, pDP1rs (PlasmidFactory, Bielefeld, Germany). Recombinant AAV1 was harvested 72 h after transfection by three cycles of freezing and thawing. The crude viral lysate was then purified by fractionation with iodixanol-gradient centrifugation. Viral genome copy titers were determined by quantitative PCR (Stratagene Mx3005p system, Agilent Technologies, Santa Clara, CA) using probes specific to the left inverted terminal repeat sequence of the AAV vector. Viral vectors were prepared by Emory viral core laboratory and the titer of AAV1 vectors was between $1.0 \times 10^{13}$ and $1.5 \times 10^{13}$ genome copies/ml.

**Animal**. WT (129S1/SvImJ mice) and $Kcne1^{-/-}$ mice (129-$Kcne1^{tm1Sfh}$/J mice) were purchased from Jackson Laboratories (California, USA). All animal-use protocols were approved by the Emory IACUC. Breeding of WT and $Kcne1^{-/-}$ mice (either sex) and genotyping procedures were the same as those described in our previous study[10]. In brief, WT and $Kcne1^{-/-}$ mice were housed with water and food available ad libitum under a 12-h light/dark cycle at a room temperature of $22 \pm 1$ °C. Common primer 5′-GAGTTCATAATGGCTGG-3′, WT primer 5′-ATGCCTGTAAACT GACC-3′, and mutant primer 5′-CCCGCTTCCATTGCTCA-3′ were used for genotyping. Mice were divided into the following five groups ($n \geq 6$ in each): (1) WT littermate controls; (2) littermate WT mice injected with the AAV1-CB7-GFP or AAV1-CB7-$Kcne1$ virus into the PSCC; (3 and 4) $Kcne1^{-/-}$ mice injected with either a low (0.5–1.0 μL, $1.5 \times 10^{13}$ genome copies/ml) or a high (1.5–2.0 μL, $1.5 \times 10^{13}$ genome copies/ml) dose of AAV1-CB7-$Kcne1$; and (5) $Kcne1^{-/-}$ mice without treatment. Left PSCC were injected and the other ear in the same mouse was used as an untreated internal control for either morphological or functional examinations.

**Surgery and viral injection procedures**. Neonatal mice were placed on ice for anesthetization. A curved incision was made in the skin behind the left ear to expose the PSCC (Supplementary Video 1). Viral solution (low dose: 0.5–1.0 μL, $1.5 \times 10^{13}$ genome copies/ml; high dose: 1.5–2.0 μL, $1.5 \times 10^{13}$ genome copies/ml) was injected using a glass micropipette with a tip size of 10–15 μm. The injection

was controlled using a Picospritzer III pressure microinjection system (Picospritzer III, Parker Hannifin Corporation, USA)[10]. The movement of glass micropipettes was controlled by a micromanipulator (MP-285, Sutter Instrument, Novato, CA) to penetrate into the PSCC through the soft bony shell (Fig. S1C). After injections, mice were placed on a 37 °C heating pad (model TR-100, Fine Science Tool Inc., Foster City, CA) to recover. Each injection took about 15–20 min to complete. More details of surgical procedures are described in our previous studies[10]. The details of surgical procedures for adult mice are described in Guo et al.[32].

**Examination of inner ear morphology**. Cardiac perfusion was performed after anesthetizing mice with ketamine hydrochloride (80 mg/Kg), xylazine (10 mg/Kg), and acepromazine (30 mg/Kg). After removal of the temporal bones, the inner ear samples—including both the cochlea and vestibular organs—were fixed in 4% paraformaldehyde overnight at 4 °C. The adult inner ear samples were decalcified in 10% EDTA for 10 days at 4 °C, and the cochleae were post-fixed with 1% osmium tetroxide, after which they were serially dehydrated in 30, 50, 70, 95, and 100% ethanol. These samples were embedded in epoxy resin (Ted Pella Inc., Redding, CA). Detailed protocols of inner ear resin sectioning and hematoxylin–eosin staining have been described in our published paper[10]. The outer diameter of the bony SSCC of some samples and the cross-sectional areas of LSCC, SSCC, and PSCC were measured by using the Photoshop software cc2018.

**Immunolabeling for *GFP* and *Kcne1* expression levels**. Whole-mount preparations of the cochleae and vestibular end organs were made for immunofluorescent labeling. Dissected samples were permeabilized in 0.2% triton in phosphate-buffered saline (PBS) for 30 min, after which they were blocked in 10% goat serum in PBS for 1 h. Primary antibodies against either GFP (1:200; Thermo Fisher, MA5-15256) or Kcne1 (1:50; Mybiosource, MBS8503082) were labeled at 4 °C overnight. After washing three times in PBS, samples were incubated with Alexa 488-conjugated secondary antibody (1:500; Invitrogen, cat# A11034). Cell membranes and cilia were stained with isothiocyanate-conjugated phalloidin (1:1000; Sigma-Aldrich, cat#P1951). Some cochlear HC bodies were labeled with the antibody against Myo7a (1:200; Proteus bioscience, cat#26–6790) and some SGNs tissues were labeled with the antibody against NF200 (1:200; Invitrogen, cat#13–1300) at 4 °C overnight. After washing three times in PBS, samples were incubated with Alexa 555-conjugated secondary antibody (1:500; Invitrogen, cat#A32727) for 1 h at room temperature. Samples were then mounted in fluoromount-G antifading solution and examined using a Zeiss LSM 510 confocal microscope. More details of the immunolabeling protocol are provided in our published paper[10].

**Measurement of ABRs and vestibular functions**. A brief protocol of our experimental design for functional studies is shown in Supplementary Fig. 1. After anesthetization with a mixture of ketamine hydrochloride (80 mg/Kg), xylazine (10 mg/Kg), and acepromazine (30 mg/Kg), we measured ABRs at 4–32 kHz. Details of the ABR testing method have been described previously[10]. Briefly, the reference electrode was inserted under the skin of the testing ear, the recording electrode was inserted under the mesial top of the skull, and the ground electrode under the skin of the contralateral ear. Tone bursts (10 ms in duration and a rise–fall time of 0.5 ms) at five frequencies (4, 8, 12, 18, 24, and 32 kHz) were generated by a Tucker–Davis system III hardware and software (TDT, Alachua, FL). The ABR threshold was measured visually based on the appearance of wave II in a series of repeatable ABR waves. The lowest sound pressure that elicited a repeatable waveform was considered to be the ABR threshold. Data acquisition and signal averaging were accomplished by the BioSigRZ software. To test vestibular functions, we recorded circling behavior, time remaining on the rotarod, and scores of swimming tests for WT and $Kcne1^{-/-}$ mice with or without treatments. Details of the vestibular functional tests can be found in previous studies[4,33]. Briefly, in the circling behavior test, each mouse was placed into an empty breeding box and allowed to acclimate to the new environment for 2 min. In the next 2 min, mouse behavior was recorded using a video camera and rotations every 2 min were counted manually. In the rotarod test, mice were placed initially on the static rod of a rotarod apparatus (Ajanta Inc., India). The time that the mouse could remain on rotating rod was recorded for the maximum duration of 120 s. The test was carried out at two different speeds, 5 rpm ($0.4 \times 10^{-3} \times g$) and 20 rpm ($6.7 \times 10^{-3} \times g$), respectively. Tests were repeated three times at each speed and the averaged duration was taken as the final result. In the swimming test, each mouse was placed into the water and swimming behavior was recorded using a video camera. The swimming behavior was scored 0–3 according to a method introduced by Isgrig et al.[4]. After each test, the mouse was dried with a towel and put into a cage on a warm heating pad. Example videos of testing during circling and swimming behaviors are provided in Supplementary Materials.

**Animal breeding and measurement of survival/growth rates**. Female and male mice were housed separately, and WT and $Kcne1^{-/-}$ mice (with or without treatment) were weighed by an electronic scale at 4, 6, and 8 weeks. Litter size, birth, and survival rates were calculated from breeding pairs among WT and $Kcne1^{-/-}$ mice with or without treatment. Breeding pairs were housed together at 8 weeks of age for 6 months, and mouse litters born to each pair during the 6 months were counted as birth rate. To determine litter survival, litters for each experimental condition were classified as survival if any of the pups survived to a weaning age of P21[12].

**Statistical analysis**. Statistical analysis was performed using the GraphPad Prism 5.0 software. All data are shown as the mean or mean ± standard error of mean ($n \pm$ SEM), and all experiments were repeated at least three times. Two-sided Student's $t$ tests were used to analyze ABR thresholds, amplitude of peak I wave, thickness of SV, cross-sections of the semicircular canals, outer diameters of bony SSCC and circling behaviors, and rotarod and swim test results, as well as secondary outcomes, including body weight growth, offspring birth, and survival rates. A $p < 0.05$ was considered to indicate a statistically significant difference.

**Reporting summary**. Further information on research design is available in the Nature Research Reporting Summary linked to this article.

## Data availability
All relevant data in the manuscript and the Supplementary Materials are available. Requests for materials should be addressed to X.L. (xlin2@emory.edu). Source data are provided with this paper.

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

## Acknowledgements
This study was supported by a grant to X.L. from the National Institute on Deafness and other Communication Disorders (RO1 DC014496).

## Author contributions
X.W. conducted and designed part of the experiments, analyzed a subset of data, and wrote part of the paper. L.Z., Y.L., W.Z., J.W., and C.C. conducted part of the experiments and analyzed a subset of data. X.L. conducted and designed all the experiments, analyzed data, and wrote the paper.

## Competing interests
The authors declare no competing interests. The authors alone are responsible for the content and writing of the paper.
