## [Peer Review File · Nature Communications]

Reviewers' Comments:

Reviewer #1:

Remarks to the Author:

The study was designed to use AAV-mediated gene therapy to treat a syndromic genetic deafness, JLNS2, which is manifested as congenital and profound hearing loss. AAV1 was used to deliver a Kcne1 gene to the Kcne1- mouse model and showed rescue in hearing and vestibular function. The study further showed as the result of the delivery, growth rate and breeding were normalized. The study is proof-of-concept that AAV gene therapy could be developed as potential treatment of JLNS in humans.

The data of the study are generally of high quality. There are some issues with the study that need to be addressed, to make the data more relevant for potential human application.

In humans JLNS2 is manifested as congenital and profound deafness. In contrast to the mouse inner ear, newborn human inner ear is fully mature. To have implication for the current strategy for human application, one needs to establish if later intervention is possible, for either hearing loss or vestibular dysfunction, preferably in mature mouse cochlea. However, recognizing the defects in adult cochlea, intervention could be tried in P7 and P14. Even the result is negative it will not diminish the significance of the current study and will provide a crucial piece of information regarding potential of the current approach in clinic.

AAV tropism depends on the cell types and maturation status. There is a major difference in infectability for an AAV in neonatal and adult mouse inner ear. For AAV1 mediated delivery, it has to be established if the infection pattern persists at later stage (P14 and P30) as in neonatal stage. If AAV1 infects mature MC and dark cells, it would support its potential utility in humans. If not, we will be back to square one finding an AAV that can infect the relevant cell types in mature inner ear.

The authors discussed extensively about the utility of the approach to treat balance problem. First of all, genetic hearing loss with balance problem is not common. Further in humans, balance problem can be generally compensated for by the visual system. The underlying causes for a majority of balance problem are unknown, and the current approach is unlikely to be useful in treating most patients with balance problems. They should tone down their claims.

Mis-expression of genes in cell types in which the genes are not supposed to be expressed could lead to unforeseen issues downstream. Could it be the decline in hearing over time may be due to mis-expression of Kcne1 in hair cells?

Fig. S4, is the damage in the Kcne1- mice uniform across all cochlear turns or limited to some regions? They should use representative regional images to illustrate them. Phalloidin labeling couldn't reveal if hair cells are still there, which should be shown by the labeling of a hair cell marker such as MYO7A.

Line 131, "Consistent with these results, the treatment apparently " What does it mean by normal size of vestibular membrane labyrinth, thickness, dimension or something else?

They should show the recovery rate comparing to WT across different frequencies, which is a good way to show how robust the recovery is.

In the legend of Fig. 5B, it stated the plot in the high-dose group came from the best 8 animals. In the text, it said it's from 20 animals. Which is which? Also, why is hearing in 8 injected animals was better than other injected animals? In any case the data from 20 injected animals has to be used and presented.

They need to study the inner ear of the mice with balance but not hearing improvement and determine if the infection pattern (e.g. only the vestibular organs were infected) as well as

morphological changes could account for the difference. The information would be very valuable in the future development of the therapy.

The growth retardation and weight loss may not only be a direct effect of Kcne1 defects. If the mothers are Kcne1-null it's likely they weren't be able to hear the vocalization of the pups and thus were unable to nurse them properly, which could contribute to the observation. One way to determine the point is to have Kcne1- pups nursed by WT mothers and see if the issue persists. Further, the pups won't be able to hear or find the mothers, which could contribute to the weight loss, which is supported that by 8 weeks, the weight differences disappeared, indicating that Kcne1- doesn't pose a long term growth problem.

Line 254, "These results indicate ...", we cannot conclude that the survival rate of Kcne1- pups was due to the balance problem based on the evidence. Much work is needed to demonstrate the point.

Line 301, "These results, for the first time, ..". This is a reason why it's important to study if hearing or balance or both can be rescued and to what degree by later intervention. The rescue requirement may be different for hearing and balance, the later intervention may still be effective for one of them (more likely balance).

The routes of injection have been previously studied. A key issue is the AAV1 infection pattern in relation to the maturation status of the cochlea. If AAV1 is no longer effective targeting mature MC or dark cells, we cannot draw any conclusion that current route of injection could be applied to clinic. In that sense the current study is not much different from WR injection mediated gene therapy.

The difference in hearing rescue of different models (Cx30 and Whirlin) is more likely due to the genes targeted and progression of hearing loss than the injection route.

Despite the early intervention, efficient transduction of relevant cell types and good hearing recovery initially, hearing recovery declined over time. What are the possible explanations?

In acknowledgement, who are the reviewers they want to thank?

Reviewer #2:

Remarks to the Author:

The authors report on gene delivery by AAV via the canalostomy route to treat hearing loss in a mouse model of JLNS, which is a sensorineural hearing loss most often caused in humans by defects on both alleles of the KCNQ1 voltage-dependent potassium channel alpha subunit or its ancillary subunit in the hear, KCNE1. KCNQ1-KCNE1 channels regulate potassium secretion into the endolymph of the inner ear and their genetic disruption causes morphological defects and profound hearing loss. The authors demonstrate convincingly that restoration of Kcne1 by AAV ameliorates the hearing loss, balance problems and morphological changes for several months after delivery. The work is somewhat novel as their previous paper focusing on the alpha subunit itself, Kcnq1, used a somewhat different delivery method. I have several specific comments.

- 1) There is no reference to Fig 2A or 2B in the text (these are the marginal cell data).
- 2) For the figures showing morphology, many do not include n values or quantitative assessment. the authors should provide in the legend an indication of how many mice the data were representative of, and where possible quantitative metrics.
- 3) Avoid use of terms such as "significant increases" (line 247). If the authors are referring to statistical significance, it is better to instead state solely "increases" and add n number, P value

and effect size afterward so that readers can assess for themselves.

4) The English needs improving throughout the manuscript.

Reviewers' comments:

Reviewer #1 (Remarks to the Author):

The study was designed to use AAV-mediated gene therapy to treat a syndromic genetic deafness, JLNS2, which is manifested as congenital and profound hearing loss. AAV1 was used to deliver a *Kcne1* gene to the *Kcne1*- mouse model and showed rescue in hearing and vestibular function. The study further showed as the result of the delivery, growth rate and breeding were normalized. The study is proof-of-concept that AAV gene therapy could be developed as potential treatment of JLNS in humans.

The data of the study are generally of high quality. There are some issues with the study that need to be addressed, to make the data more relevant for potential human application.

In humans JLNS2 is manifested as congenital and profound deafness. In contrast to the mouse inner ear, newborn human inner ear is fully mature. To have implication for the current strategy for human application, one needs to establish if later intervention is possible, for either hearing loss or vestibular dysfunction, preferably in mature mouse cochlea. However, recognizing the defects in adult cochlea, intervention could be tried in P7 and P14. Even the result is negative it will not diminish the significance of the current study and will provide a crucial piece of information regarding potential of the current approach in clinic.

AAV tropism depends on the cell types and maturation status. There is a major difference in infectability for an AAV in neonatal and adult mouse inner ear. For AAV1 mediated delivery, it has to be established if the infection pattern persists at later stage (P14 and P30) as in neonatal stage. If AAV1 infects mature MC and dark cells, it would support its potential utility in humans. If not, we will be back to square one finding an AAV that can infect the relevant cell types in mature inner ear.

The authors discussed extensively about the utility of the approach to treat balance problem. First of all, genetic hearing loss with balance problem is not common. Further in humans, balance problem can be generally compensated for by the visual system. The underlying causes for a majority of balance problem are unknown, and the current approach is unlikely to be useful in treating most patients with balance problems. They should tone down their claims.

Mis-expression of genes in cell types in which the genes are not supposed to be expressed could lead to unforeseen issues downstream. Could it be the decline in hearing over time may be due to mis-expression of *Kcne1* in hair cells?

Fig. S4, is the damage in the *Kcne1*- mice uniform across all cochlear turns or limited to some regions? They should use representative regional images to illustrate them. Phalloidin labeling couldn't reveal if hair cells are still there, which should be shown by the labeling of a hair cell marker such as *MYO7A*. Line 131, "Consistent with these results, the treatment apparently " What does it mean by normal size of vestibular membrane labyrinth, thickness, dimension or something else? They should show the recovery rate comparing to WT across different frequencies, which is a good way to show how robust the recovery is.

In the legend of Fig. 5B, it stated the plot in the high-dose group came from the best 8 animals. In the text, it said it's from 20 animals. Which is which? Also, why is hearing in 8 injected animals was better than other injected animals? In any case the data from 20 injected animals has to be used and presented.

They need to study the inner ear of the mice with balance but not hearing improvement and determine if the infection pattern (e.g. only the vestibular organs were infected) as well as morphological changes could account for the difference. The information would be very valuable in the future development of the therapy.

The growth retardation and weight loss may not only be a direct effect of *Kcne1* defects. If the mothers are *Kcne1*-null it's likely they weren't be able to hear the vocalization of the pups and thus were unable to nurse them properly, which could contribute to the observation. One way to determine the point is to have *Kcne1*- pups nursed by WT mothers and see if the issue persists. Further, the pups won't be able to hear or find the mothers, which could contribute to the weight loss, which is supported that by 8 weeks, the weight differences disappeared, indicating that *Kcne1*- doesn't pose a long term growth problem.

Line 254, "These results indicate ...", we cannot conclude that the survival rate of *Kcne1*- pups was due to the balance problem based on the evidence. Much work is needed to demonstrate the point.

Line 301, "These results, for the first time, ..". This is a reason why it's important to study if hearing or balance or both can be rescued and to what degree by later intervention. The rescue requirement may be different for hearing and balance, the later intervention may still be effective for one of them (more likely balance).

The routes of injection have been previously studied. A key issue is the AAV1 infection pattern in relation to the maturation status of the cochlea. If AAV1 is no longer effective targeting mature MC or dark cells, we cannot draw any conclusion that current route of injection could be applied to clinic. In that sense the current study is not much different from WR injection mediated gene therapy.

The difference in hearing rescue of different models (*Cx30* and *Whirlin*) is more likely due to the genes targeted and progression of hearing loss than the injection route.

Despite the early intervention, efficient transduction of relevant cell types and good hearing recovery initially, hearing recovery declined over time. What are the possible explanations?

In acknowledgement, who are the reviewers they want to thank?

Reviewer #2 (Remarks to the Author):

The authors report on gene delivery by AAV via the canalostomy route to treat hearing loss in a mouse model of JLNS, which is a sensorineural hearing loss most often caused in humans by defects on both alleles of the *KCNQ1* voltage-dependent potassium channel alpha subunit or its ancillary subunit in the hear, *KCNE1*. *KCNQ1-KCNE1* channels regulate potassium secretion into the endolymph of the inner ear and their genetic disruption causes morphological defects and profound hearing loss. The authors demonstrate convincingly that restoration of *Kcne1* by AAV ameliorates the hearing loss, balance problems and morphological changes for several months after delivery. The work is somewhat novel as their previous paper focusing on the alpha subunit itself, *Kcnq1*, used a somewhat different delivery method. I have several specific comments.

1) There is no reference to Fig 2A or 2B in the text (these are the marginal cell data).

2) For the figures showing morphology, many do not include n values or quantitative assessment. the authors should provide in the legend an indication of how many mice the data were representative of,

and where possible quantitative metrics.

3) Avoid use of terms such as "significant increases" (line 247). If the authors are referring to statistical significance, it is better to instead state solely "increases" and add n number, P value and effect size afterward so that readers can assess for themselves.

4) The English needs improving throughout the manuscript.

Answers to Reviewers' comments:

Reviewer #1 (Remarks to the Author):

The study was designed to use AAV-mediated gene therapy to treat a syndromic genetic deafness, JLNS2, which is manifested as congenital and profound hearing loss. AAV1 was used to deliver a *Kcne1* gene to the *Kcne1*- mouse model and showed rescue in hearing and vestibular function. The study further showed as the result of the delivery, growth rate and breeding were normalized. The study is proof-of-concept that AAV gene therapy could be developed as potential treatment of JLNS in humans. The data of the study are generally of high quality. There are some issues with the study that need to be addressed, to make the data more relevant for potential human application.

ANSWER: We thank the Reviewers for helping us improve the quality of our work.

In humans JLNS2 is manifested as congenital and profound deafness. In contrast to the mouse inner ear, newborn human inner ear is fully mature. To have implication for the current strategy for human application, one needs to establish if later intervention is possible, for either hearing loss or vestibular dysfunction, preferably in mature mouse cochlea. However, recognizing the defects in adult cochlea, intervention could be tried in P7 and P14. Even the result is negative it will not diminish the significance of the current study and will provide a crucial piece of information regarding potential of the current approach in clinic.

ANSWER: We agree with the reviewer's comments that results from injections done at later developmental stages may have stronger implications for human intervention. The first published paper about the *Kcne1*^{-/-} mice (Vetter, Mann et al. 1996) show that the collapse of Reissner's membrane and degeneration of the organ of Corti across all cochlear turns happen as early as P3 (Vetter, Mann et al. 1996). Our published data indicate that gene therapy implemented after degeneration of cochlear cells fails to yield any significant treatment efficacy (Zhang, Kim et al. 2018). These results are consistent with our finding in this study that injections done at P3 had little treatment efficacy (Results given in Supplementary Fig.5). We have summarized our data suggesting that optimal treatment time window for *Kcne1*^{-/-} mice for preserving hearing and vestibular functions is between P0–P2, and interventions done at and after P3 are unsuccessful.

We have modified manuscript to clearly present these findings:

- (1) Starting from Page5 line121: “In addition, injections of high-dosage of AAV1-CB7-Kcne1 at P3 in *Kcne1*^{-/-} mice (n=4) did not yield comparable hearing improvements as those injections at P0–P2 (Supplementary Fig. 5). We therefore focused on our studies about gene therapy of the *Kcne1*^{-/-} mice to those injected between P0–P2
- (2) Starting from Page13 line297: “Following major factors were considered in our experimental designs and data analyses: (1) gene therapy implemented after degeneration of cochlear cells would likely to fail²³; (2) our injections done at P3 failed to give effective treatment results in hearing preservation (Supplementary Fig. 5); and (3) the extremely small diameter of the PSCC in untreated adult *Kcne1*^{-/-} mice made the injections done to the adult PSCC unsuccessful (Fig. 3)
- (3) We added data in Supplementary Fig. 5 to show ABR tests and vestibular behavioral assessments at P30 after a high dosage of AAV1-CB7-*Kcne1* was injected by the PSCC route at P3 for *Kcne1*^{-/-} mice. Results showed that injections done at P3 failed show efficacy.

AAV tropism depends on the cell types and maturation status. There is a major difference in infectability for an AAV in neonatal and adult mouse inner ear. For AAV1 mediated delivery, it has to be established if the infection pattern persists at later stage (P14 and P30) as in neonatal stage. If AAV1 infects mature MC and dark cells, it would support its potential utility in humans. If not, we will be back to square one finding an AAV that can infect the relevant cell types in mature inner ear.

ANSWER: Our data indicate that injections made at neonatal stage (P0–P2) successfully transduce many vestibular and cochlear cells in WT mice (Fig.1), and injections made at adult stage show the similar infection pattern but lower transduction efficiency in WT mice (Supplementary Fig. 4). However, the diameter of each of the ossified semicircular canals in adult *Kcne1*^{-/-} mice is extremely small (Fig.3), which made injections at later stages unsuccessful in *Kcne1*^{-/-} mice. In addition, our data show that even the injections done at P3 failed to yield treatment effect in hearing preservation (Supplementary Fig. 5).

We have modified manuscript accordingly and the relevant texts are copied here (starting from Page5 line114):“ When AAV1-CB7-GFP was injected later at P30 into the WT mice (n=6), many inner ear cells (e.g., vestibular dark cells, supporting cells and inner hair cells, etc.) (Supplementary Fig. 4a–c) were still transduced to express GFP at the adult stage. The ABRs, measured seven days after injections, were normal in the injected adult WT mice (n=6; Supplementary Fig. 4d), supporting that the injection procedures done at the adult stage didn’t damage hearing. Apparently due to the abnormal development of the semicircular canals in *Kcne1*^{-/-} mice that give rise to much smaller canals and degenerated vestibular membrane at the adult stage (Fig. 3), we could not successfully obtain viral transduction in the adult *Kcne1*^{-/-} mice. In addition, injections of high-dosage of AAV1-CB7-Kcne1 at P3 in *Kcne1*^{-/-} mice (n=4) did not yield comparable hearing improvements as those injections at P0–P2 (Supplementary Fig. 5). We therefore focused on our studies about gene therapy of the *Kcne1*^{-/-} mice to those injected between P0–P2.”

The authors discussed extensively about the utility of the approach to treat balance problem. First of all, genetic hearing loss with balance problem is not common. Further in humans, balance problem can be generally compensated for by the visual system. The underlying causes for a majority of balance problem

are unknown, and the current approach is unlikely to be useful in treating most patients with balance problems. They should tone down their claims.

ANSWER: We have rewritten the manuscript to include these comments made by the Reviewer (starting from page17 line382): “In summary, our results support that inner ear gene therapy using the canalostomy approach effectively preserved vestibular and hearing functions in mouse models, and the treatment efficacy for the vestibular phenotypes may be independently displayed. However, genetic hearing loss with balance dysfunction in humans is uncommon (e.g., due to compensation by the visual system). The underlying causes for a majority of clinical balance problems are unclear. We thus can’t make a conclusion that the current gene therapy approach demonstrated in the *Kcne1*^{-/-} mouse model via canalostomy approach is widely applicable to treat patients with balance problems. However, given the extent and duration of the functional recovery displayed here using morphological, physiological, and behavioral assessments, our results suggest that *Kcne1* gene replacement therapy for recessive KCNE1 mutations in human JLNS2 patients is promising for further development of cochlear and vestibular gene therapies. To advance toward clinical translational goals, future studies are also needed to investigate longer-term efficacy and immunoreactivity of AAVs in inner ear gene therapy.”

Mis-expression of genes in cell types in which the genes are not supposed to be expressed could lead to unforeseen issues downstream. Could it be the decline in hearing over time may be due to mis-expression of *Kcne1* in hair cells?

ANSWER: We agree with the reviewer’s point that mis-expression of genes in ectopic cells could lead to unforeseen issues downstream if longer period of observation time was followed.

Our results demonstrated that viral inoculation with AAV1-CB7-*Kcne1* into the PSCC reliably transduced a large percentage of cells in the inner ear targeted for treatment. *Kcne1* expressions were also found ectopically in many types of inner ear cells. However, after injecting AAV1-CB7-*Kcne1* into the PSCC of WT mice, ABR thresholds tested at P30 were unchanged on the injected side compared to the un-injected ears of the same mouse (Supplementary Fig. 2b), suggesting that ectopic *Kcne1* expression did not affect normal hearing in WT mice, at least one month after injection. We are unclear whether ectopic expression of *Kcne1* in inner ear will lead to hearing loss over a longer time period. These will need long-term follow up studies in the future.

We have amended the manuscript to include this point raised by the Reviewer:

(1) Starting from Page14 Line325: “ We showed that ectopic expression of *Kcne1* in the inner ear didn’t affect hearing threshold of WT mice tested at one month (Supplementary Fig. 2a, b), although future studies are required to determine whether ectopic expression of *Kcne1* in the inner ear (e.g., in hair cells) can gradually lead to hearing loss. ”.

(2) Starting from Page16 line374: “It is unclear for the reason of efficacy decline. Possible explanations may include a gradual decline of the virally mediated expression or excessive ectopic expression of targeted genes.”.

Fig. S4, is the damage in the *Kcne1*- mice uniform across all cochlear turns or limited to some regions? They should use representative regional images to illustrate them. Phalloidin labeling couldn’t reveal if hair cells are still there, which should be shown by the labeling of a hair cell marker such as MYO7A.

ANSWER: The collapse of Reissner's membrane and all cochlear hair cells degeneration occurred across all cochlear turns in neonatal and adult *Kcne1*^{-/-} mice. Our result is consistent with Vetter's first descriptions of degeneration patterns in the inner ear of *Kcne1*^{-/-} mice (Vetter, Mann et al. 1996). In Fig.S4 (Fig. 4 in revised manuscript), we showed the basilar membrane in the middle turn as the representative image. We agree with the reviewer that damage to hair cells can be better labeled with Myo7A. However, phalloidin labeling shows the condition of the ciliary bundles of cochlear and vestibular hair cells which is often the first indication for the condition of hair cells. We have changed the text through the manuscript to reflect this point that our studies only showed the damage of hair bundle, without assuming the condition of hair cells:

- (1) Starting from Page6 Line128: "In untreated ears (Fig. 4b) most cochlear and vestibular hair cells (HCs) were severely damaged as indicated by loss of ciliary bundles. The cell border and cell size of MCs were often irregular (Fig. 4b).
- (2) Starting from Page6 Line131: "In *Kcne1*^{-/-} mice treated with the low-dosage (n=4; Fig. 4c), the normal morphologies of ciliary bundles of cochlear IHCs and vestibular HCs in the utricle and CA appeared to be preserved,
- (3) Starting from Page6 Line136: "The ciliary bundles of cochlear HCs and vestibular HCs, as well as the hexagonal shape of the MCs in the SV, appeared to be normal in the high-dose-treated ears of *Kcne1*^{-/-} mice (Fig. 4d).
- (4) Starting from Page8 Line182: "In the low-dosage group (Supplementary Fig. 7b), only a few vestibular HCs showed normal ciliary bundles in the utricle and CA at P6m. Severe damage or degeneration was also found in

Line 131, "Consistent with these results, the treatment apparently " What does it mean by normal size of vestibular membrane labyrinth, thickness, dimension or something else?
They should show the recovery rate comparing to WT across different frequencies, which is a good way to show how robust the recovery is.

ANSWER: We measured the diameter of bony labyrinth since it is unreliable to quantify changes in the diameter of the soft vestibular membranous labyrinth, as part of the data showing the treatment effect on the correction of morphological phenotypes of *Kcne1*^{-/-} mice. The results about hearing preservation is shown in another figure (Fig. 6).

To clarify these points, we modified the text description for this part of the result (starting from Page6 Line141): "The average outer diameter of the bony superior semicircular canal (SSCC) in treated ears of *Kcne1*^{-/-} mice was almost twice as large ($256.6 \pm 8.4 \mu\text{m}$) as that of untreated ears ($114.1 \pm 8.0 \mu\text{m}$) (n=8 in each group, $p < 0.0001$, student's *t* test; Fig. 3b, d), which brought the average outer diameter of the SSCC in the treated ears of *Kcne1*^{-/-} mice similar to that of WT mice ($269.6 \pm 7.0 \mu\text{m}$, n=8 in each group, $p = 0.65$, student's *t* test; Fig. 3d). Consistent with these results, gross morphology of the vestibular membranous labyrinth after the treatment (Fig. 3c) appeared to be larger, although it was problematic to accurately quantify changes in the size of the soft vestibular membranous labyrinth. The treatments also corrected the reduction in the cavities of the three semicircular canals. "

In the legend of Fig. 5B, it stated the plot in the high-dose group came from the best 8 animals. In the text,

it said it's from 20 animals. Which is which? Also, why is hearing in 8 injected animals was better than other injected animals? In any case the data from 20 injected animals has to be used and presented.

ANSWER: Averaged ABR thresholds across all frequencies tested for 20 injected mice are given in Fig. 6b in the revised manuscript. The results included the data from 8 animals that showed the better hearing improvement and 12 other mice injected. To avoid confusion and misunderstanding, we deleted the data curve separately showing the 8 mice in the revised manuscript (Fig. 6b), corresponding text description was also deleted.

They need to study the inner ear of the mice with balance but not hearing improvement and determine if the infection pattern (e.g. only the vestibular organs were infected) as well as morphological changes could account for the difference. The information would be very valuable in the future development of the therapy.

ANSWER: 65% (13/20) of low-dose-treated mice and 20% (4/20) of high-dose-treated mice showed improvement in balance but not hearing function. Comparing to mice that showed improvement in both balance and hearing, the mice in this category showed the same general viral infection pattern in the vestibular system but the expression level was much less (Fig. 2c and Supplementary Fig. 4b in manuscript).

We amended this section by adding the following sentences (page12line243): “Comparing to the treated mice that showed improvement in both balance and hearing, the other treated mice with vestibular improvements but not hearing improvements showed the same general viral infection pattern in the vestibular dark cells and MCs but the expression level was much less (Fig. 2c). These results supported that vestibular dysfunction in *Kcne1*^{-/-} mice was improved after injecting the AAV1-CB7-*Kcne1* viral construct into the PSCC, even for mice in the low-dosage group in which only a slight or no hearing improvement was observed.”

The growth retardation and weight loss may not only be a direct effect of *Kcne1* defects. If the mothers are *Kcne1*-null it's likely they weren't be able to hear the vocalization of the pups and thus were unable to nurse them properly, which could contribute to the observation. One way to determine the point is to have *Kcne1*- pups nursed by WT mothers and see if the issue persists. Further, the pups won't be able to hear or find the mothers, which could contribute to the weight loss, which is supported that by 8 weeks, the weight differences disappeared, indicating that *Kcne1*- doesn't pose a long term growth problem.

ANSWER: We agree with the reviewer's comments that the growth retardation and weight loss may not be a direct effect of *Kcne1* null, and *Kcne1*-null may not pose a long-term growth problem. Consistent with the Reviewer's suggestions, we found that *Kcne1*^{-/-} mother mice were unable to nurse their babies properly. Survival rate was significantly increased when *Kcne1*^{-/-} baby mice were nursed by WT mothers. We modified manuscript and added additional data to clarify these points:

Starting from Page11 Line256: “We hypothesized that constant circling behavior and other vestibular dysfunctions may affect the growth, breeding productivity, and offspring survival rates among *Kcne1*^{-/-} mice, and gene therapy may alleviate these phenotypes. Compared to that of WT mice, the untreated *Kcne1*^{-/-} mice nursed by *Kcne1*^{-/-} mothers had a significantly lower body weight at four (14.2 ± 0.4 g) and six (18.3 ± 0.4 g) weeks of age (n=6, p=0.007 at 4 weeks, p=0.02 at 6 weeks, student's *t* tests). In contrast, the untreated *Kcne1*^{-/-} mice nursed by

WT mothers, and the treated *Kcne1*^{-/-} mice in both low- and high-dosage groups nursed by WT mothers, showed similar weight growth comparing to WT mice during 8-week period (n=6, p>0.05 in all comparisons, student's *t* tests; Supplementary Fig. 8a). During a six-month mating period, four WT breeding pairs produced 5-6 litters each, or ~1 litter/month. When breeding productivity was compared, we found that four WT breeding pairs produced a total of 22 litters during the six-month period, whereas four untreated *Kcne1*^{-/-} breeding pairs produced a total of 11 litters during the same period. In comparison, four high-dose-treated *Kcne1*^{-/-} breeding pairs and four low-dose-treated *Kcne1*^{-/-} breeding pairs produced 17 and 16 litters during the six-month period, respectively. ...”.

Line 254, “These results indicate ...”, we cannot conclude that the survival rate of *Kcne1*- pups was due to the balance problem based on the evidence. Much work is needed to demonstrate the point.

ANSWER: We agree with reviewer's comment and added these sentences (Page12 Line281): “..... . The reason for these improvements is unclear, although we speculate that better vestibular functions may play a role, however much work remains to be done to clarify the underlying mechanism.”

Line 301, “These results, for the first time, ..”. This is a reason why it's important to study if hearing or balance or both can be rescued and to what degree by later intervention. The rescue requirement may be different for hearing and balance, the later intervention may still be effective for one of them (more likely balance).

ANSWER: We agree with the reviewer's comments and we have extensively modified our manuscript to include these suggested points. In our study, and consistent with the first report of *Kcne1*^{-/-} mice by Vetter et al. (see citation#20), we found that degeneration of inner ear cells started at P3. These findings may explain why our injections performed at P3 failed to yield positive treatment effects. In addition, the diameter of the ossified semicircular canals in *Kcne1*^{-/-} mice was extremely smaller at P30 (data given in Fig.3).

We added following sections to clarify these points:

(1) Starting from Page5 Line119: “ Apparently due to the abnormal development of the semicircular canals in *Kcne1*^{-/-} mice that give rise to much smaller canals and degenerated vestibular membrane at the adult stage (Fig. 3), we could not successfully obtain viral transduction in the adult *Kcne1*^{-/-} mice. In addition, injections of high-dosage of AAV1-CB7-*Kcne1* at P3 in *Kcne1*^{-/-} mice (n=4) did not yield comparable hearing improvements as those injections at P0–P2 (Supplementary Fig. 5). We therefore focused on our studies about gene therapy of the *Kcne1*^{-/-} mice to those injected between P0–P2. ”.

(2) Starting from Page13 Line297 (in Discussion section): “Following major factors were considered in our experimental designs and data analyses: (1) gene therapy implemented after degeneration of cochlear cells would likely to fail²³; (2) our injections done at P3 failed to give effective treatment results in hearing preservation (Supplementary Fig. 5); and (3) the extremely small diameter of the PSCC in untreated adult *Kcne1*^{-/-} mice made the injections done to the adult PSCC unsuccessful (Fig. 3). We therefore focused analyzing treatment results to those performed in a time window between P0–P2. ”.

The routes of injection have been previously studied. A key issue is the AAV1 infection pattern in relation to the maturation status of the cochlea. If AAV1 is no longer effective targeting mature MC or dark cells, we cannot draw any conclusion that current route of injection could be applied to clinic. In that sense the current study is not much different from WR injection mediated gene therapy.

ANSWER: We provided AAV1-mediated GFP expression data, which were obtained by delivering via the canalostomy approach in the adult (at P30) WT mice. Results obtained at the adult stage showed that AAV1 successfully transduced vestibular cells (e.g., dark cells, supporting cells) and cochlear inner hair cells but not MCs (Supplementary Fig. 4).

Following sentences were add or modified to clarify these issues (starting from Page5 Line114): “When AAV1-CB7-GFP was injected later at P30 into the WT mice (n=6), many inner ear cells (e.g., vestibular dark cells, supporting cells and inner hair cells, etc.) (Supplementary Fig. 4a–c) were still transduced to express GFP at the adult stage. The ABRs, measured seven days after injections, were normal in the injected adult WT mice (n=6; Supplementary Fig. 4d), supporting that the injection procedures done at the adult stage didn’t damage hearing.”

The difference in hearing rescue of different models (Cx30 and Whirlin) is more likely due to the genes targeted and progression of hearing loss than the injection route.

ANSWER: We agree with the reviewer’s comment that the difference in hearing rescue of different models (Cx30, Whirlin and Kcne1) is likely due to the genes targeted and differences in the progression of hearing loss. We modified relevant sections in the Discussion (Page16 Line368): “These differences in the degrees of hearing improvements in different models (*Kcne1*^{-/-}, *Cx30*^{Δ/Δ}, *Cx30*^{-/-} and *Whirlin*^{-/-}) are likely due to differences in genes targeted and the time courses of morphological and hearing deteriorations associated with each gene.”

Despite the early intervention, efficient transduction of relevant cell types and good hearing recovery initially, hearing recovery declined over time. What are the possible explanations?

ANSWER: We added following sentences in the Discussion to address this issue:

(1) Starting from Page14 Line325: “We showed that ectopic expression of *Kcne1* in the inner ear didn’t affect hearing threshold of WT mice tested at one month (Supplementary Fig. 2a, b), although future studies are required to determine whether ectopic expression of *Kcne1* in the inner ear (e.g., in hair cells) can gradually lead to hearing loss.”

(2) Starting from Page16 Line371: “One common finding among published studies is in the transient nature of treatment efficacies. Even in the high-dosage group of mice in our study, we observed that hearing improvements started to decline at five months after treatment. In Isgrig et al.⁴, this decline occurred at less than four months after treatment. It is unclear for the reason of efficacy decline. Possible explanations may include a gradual decline of the virally mediated expression or excessive ectopic expression of targeted genes.”

(3) Starting from Page17 Line 393: “To advance toward clinical translational goals, future studies are also needed to investigate longer-term efficacy and immunoreactivity of AAVs in inner ear gene therapy.”

In acknowledgement, who are the reviewers they want to thank?

ANSWER: We deleted this sentence since this was obtained from a paid commercial service company which provided English proofreading.

Reviewer #2 (Remarks to the Author):

The authors report on gene delivery by AAV via the canalostomy route to treat hearing loss in a mouse model of JLNS, which is a sensorineural hearing loss most often caused in humans by defects on both alleles of the KCNQ1 voltage-dependent potassium channel alpha subunit or its ancillary subunit in the hear, KCNE1. KCNQ1-KCNE1 channels regulate potassium secretion into the endolymph of the inner ear and their genetic disruption causes morphological defects and profound hearing loss. The authors demonstrate convincingly that restoration of *Kcne1* by AAV ameliorates the hearing loss, balance problems and morphological changes for several months after delivery. The work is somewhat novel as their previous paper focusing on the alpha subunit itself, *Kcnq1*, used a somewhat different delivery method. I have several specific comments.

ANSWER: We thank the reviewer for their appreciation of the quality of our work. We are also very grateful for the reviewer's help in improving the quality of manuscript.

1) There is no reference to Fig 2A or 2B in the text (these are the marginal cell data).

ANSWER: Fig. 2a and Fig. 2b are described in the texts in these locations in the revised manuscript:

(1) Starting from Page4 Line98: “*Kcne1* is normally expressed on the apical membrane of MCs and vestibular dark cells (Fig. 2a)^{14,15}. ”.

(2) Starting from Page4 Line99: “Consistent with the expected results from the *Kcne1*^{-/-} mice, no *Kcne1* expression was detected in the MCs and vestibular dark cells of untreated *Kcne1*^{-/-} mice (Fig. 2b). ”.

2) For the figures showing morphology, many do not include n values or quantitative assessment. the authors should provide in the legend an indication of how many mice the data were representative of, and where possible quantitative metrics.

ANSWER: We have re-written legends for these figures in the revised manuscript to include information suggested by the Reviewer.

3) Avoid use of terms such as "significant increases" (line 247). If the authors are referring to statistical significance, it is better to instead state solely "increases" and add n number, P value and effect size afterward so that readers can assess for themselves.

ANSWER: In the revised and throughout the manuscript, we only used “significant increase” when *p* value is provided for the results.

4) The English needs improving throughout the manuscript.

ANSWER: We have carefully revised the manuscript. This manuscript has been proofread and edited by a professional editing service company, LetPub (www.letpub.com).

Reviewers' Comments:

Reviewer #1:

Remarks to the Author:

Comments

The revised manuscript addressed most of the issues raised. There are a few points that need to be further addressed.

They need to show the cell types AAV1-GFP transduced in vestibule and ganglions by specific markers. Phalloidin labeling does not provide cell identity information critical to the understanding of the recovery offered by gene therapy.

In their rebuttal letter, they stated that AAV1 showed a similar infection pattern in adult as in neonatal stage. However in Supplementary Fig. 4b there is no transduction by AAV in the MC in adult. This raises doubt about its utility in humans. This needs to be discussed in the discussion.

2 out of 6 mice at 6 months showed some degree of preservation in hair cells and other cell types. What about 4 others? Please provide the counting data to show survival. This data should be used to correlate with hearing test results at 6 months. If cell survival protection is minimal, it should be discussed why they saw a reasonable hearing rescue at 6 months despite the lack of preservation.

In the rebuttal, they mentioned that when *Kcne1*(-/-) pups were nursed by a WT mother, the abnormalities they observed went away. The data need to be presented.

First sentence in page 18, it is overstated, as the issues discussed are only related to mice. Should change the sentence to describe the relevance in mice.

In the discussion the potential issues need to be discussed that include 1). Identify AAV that can infect adult MC; 2). The need to address if a window of opportunity for intervention still exists in human due to rapid degeneration of relevant cells; 3) If the injection route is still available in human patients if the inner ear structure is severely damaged.

Please use * to indicate significance in the figures, as some of them do not show significant changes.

Line 20 p2, "Results showed the treatment" should be "Results showed early treatment"

Line 23, p2, "(16 out of 20 mice, or 16/20) " should be "(16 out of 20 mice)"

Reviewer #2:

Remarks to the Author:

The authors have addressed most of my comments satisfactorily. However, they have not appropriately addressed the following:

"3) Avoid use of terms such as "significant increases" (line 247). If the authors are referring to statistical significance, it is better to instead state solely "increases" and add n number, P value and effect size afterward so that readers can assess for themselves.

ANSWER: In the revised and throughout the manuscript, we only used "significant increase" when p value is provided for the results."

This is the opposite of what I was suggesting. Many prominent statisticians are advising that the use of the term "significant" be AVOIDED COMPLETELY in this context because it is essentially

meaningless. Please remove the word and instead describe the effect, e.g., X was increased xx-fold ($P = xx$; $n = xx$). Also the authors still use the term "significant" in many other parts of the manuscript where they do not quote P values, such as in the abstract and introduction. The term really is misleading.

Reviewers' original comments are copied here:

Reviewer #1 (Remarks to the Author):

Comments

The revised manuscript addressed most of the issues raised. There are a few points that need to be further addressed.

They need to show the cell types AAV1-GFP transduced in vestibule and ganglions by specific markers. Phalloidin labeling does not provide cell identity information critical to the understanding of the recovery offered by gene therapy.

In their rebuttal letter, they stated that AAV1 showed a similar infection pattern in adult as in neonatal stage. However in Supplementary Fig. 4b there is no transduction by AAV in the MC in adult. This raises doubt about its utility in humans. This needs to be discussed in the discussion.

2 out of 6 mice at 6 months showed some degree of preservation in hair cells and other cell types. What about 4 others? Please provide the counting data to show survival. This data should be used to correlate with hearing test results at 6 months. If cell survival protection is minimal, it should be discussed why they saw a reasonable hearing rescue at 6 months despite the lack of preservation.

In the rebuttal, they mentioned that when *Kcne1*(-/-) pups were nursed by a WT mother, the abnormalities they observed went away. The data need to be presented.

First sentence in page 18, it is overstated, as the issues discussed are only related to mice. Should change the sentence to describe the relevance in mice.

In the discussion the potential issues need to be discussed that include 1). Identify AAV that can infect adult MC; 2). The need to address if a window of opportunity for intervention still exists in human due to rapid degeneration of relevant cells; 3) If the injection route is still available in human patients if the inner ear structure is severely damaged.

Please use * to indicate significance in the figures, as some of them do not show significant changes.

Line 20 p2, "Results showed the treatment" should be "Results showed early treatment"

Line 23, p2, "(16 out of 20 mice, or 16/20) " should be "(16 out of 20 mice)"

Reviewer #2 (Remarks to the Author):

The authors have addressed most of my comments satisfactorily. However, they have not appropriately addressed the following:

"3) Avoid use of terms such as "significant increases" (line 247). If the authors are referring

to
statistical significance, it is better to instead state solely "increases" and add n number, P
value
and effect size afterward so that readers can assess for themselves.

This is the opposite of what I was suggesting. Many prominent statisticians are advising that the use of the term "significant" be AVOIDED COMPLETELY in this context because it is essentially meaningless. Please remove the word and instead describe the effect, e.g., X was increased xx-fold (P = xx; n = xx). Also the authors still use the term "significant" in many other parts of the manuscript where they do not quote P values, such as in the abstract and introduction. The term really is misleading.

Reviewers' comments and point-to-point answers

Reviewer #1 (Remarks to the Author):

Comments

The revised manuscript addressed most of the issues raised. There are a few points that need to be further addressed.

They need to show the cell types AAV1-GFP transduced in vestibule and ganglions by specific markers. Phalloidin labeling does not provide cell identity information critical to the understanding of the recovery offered by gene therapy.

ANSWER: We agree with the reviewer that phalloidin labeling gives the outline of cells and only is helpful to a limited degree in identifying the cell identity when combined with location information, and such method does not provide a definitive answer to cell identity for cochlear spiral ganglion neurons (SGNs) and vestibular HCs. As suggested, we added new data (Fig.1m, n; and Fig.1o-r) showing that many GFP-positive cells resulting from AAV1-CB7-GFP transduction were also labeled with the antibody against either NF200 (Fig.1m, n) or Myo7a (Fig.1o-r) respectively, supporting that the viral construct we used was capable of transducing SGNs and vestibular HCs. We also added new data in supplementary Fig.6, in which morphological improvement of cochlear HCs was examined with labeling of Myo7A antibody which is specific for the HCs. We believe these new data helped us to answer the Reviewer's question about understanding the recovery effects offered by gene therapy conducted in this study.

We modified following texts accordingly to include the new data (starting from Page4 line84): "GFP-positive cells were also found in SGNs regions (Fig. 1a, b) and vestibular compartments (Fig. 1g-l). Extensive GFP signals were confirmed to be in the SGNs (Fig. 1m, n) by specific labeling with antibody against NF200, and in vestibular HCs which were identified by specific labeling with antibody against

Myosin 7a (*Myo7a*) (Fig. 1o–r). GFP-positive cells were also found in apparent supporting cells in the vestibular compartments including the saccule (Fig. 1g, h, o, p), utricle (Fig. 1i, j, q, r) and crista ampullaris (CA) (Fig. 1k, l).”

Starting from Page6 line143: “The bodies of cochlear HCs were labeled with an antibody against *Myo7a* to further observe the morphological changes of cochlear HCs after AAV1-CB7-*Kcne1* injection. In WT cochleae, the shape and arrangement of IHCs and OHCs displayed their normal patterns (Supplementary Fig. 6a–c). However, both types of cochlear HCs were severely degenerated in all turns in untreated ears of *Kcne1*^{-/-} mice (Supplementary Fig. 6d–f). In low-dose-treated ears of *Kcne1*^{-/-} mice (Supplementary Fig. 6g–i), the number and shape of IHCs were apparently normal in all turns. However, most OHCs appeared to be degenerated in the middle and basal turns. After injections with the high-dosage of AAV1-CB7-*Kcne1*, IHCs and OHCs in all cochlear turns appeared to be normal (Supplementary Fig. 6j–l). These results were consistent with the morphological changes we observed with labeling of ciliary bundles of cochlear HCs by phalloidin (Fig. 4).”

Thus, the therapeutic effects we observed for AAV1-*Kcne1* gene therapy were examined and supported by results from the following four independent lines of observations:

(1) Gross morphology observations of organ development: the outer diameters of bony semicircle canals and vestibular membranous labyrinth;

(2) Observations made from whole-mount preparations: ciliary bundles and cell bodies of cochlear HCs, ciliary bundles of vestibular HCs, morphology (hexagonal shapes) of marginal cells in the SV;

(3) Morphological shape and measurements made from resin sections: the cross-sectional areas of cavities of three semicircular canals, the locations of the Reissner’s membrane, the thickness of the SV, the general development of the organ of Corti and SGNs;

(4) Hearing and vestibular function assessments.

At all four levels we observed consistent results supporting therapeutic efficacy.

In their rebuttal letter, they stated that AAV1 showed a similar infection pattern in adult as in neonatal stage. However in Supplementary Fig. 4b there is no transduction by AAV in the MC in adult. This raises doubt about its utility in humans. This needs to be discussed in the discussion.

ANSWER: When AAV1-CB7-*GFP* was injected at P30 into the WT mice, many vestibular dark cells, supporting cells and inner ear cells (Supplementary Fig. 4a–c) were successfully transduced to express GFP, but MCs in the SV were negative. We agree with the Reviewer that the negative transduction in adult MCs needs to be discussed for its implications in further application in human translation.

We have added following text:

Starting from Page5 line117: “When AAV1-CB7-*GFP* was injected later at

P30 into the WT mice (n=6), many inner ear cells (e.g., vestibular dark cells, supporting cells and IHCs) were still transduced at the adult stage, although the MCs in the SV were not transduced at the adult stage suggesting adult treatment may not be successful.”

Starting from Page17 line399: “Considering that AAV1-CB7-GFP could not transduce the adult MCs of mice in the present study, it seems to be necessary to identify new serotypes of AAVs or other means to transduce adult MCs in future human translational studies.”

2 out of 6 mice at 6 months showed some degree of preservation in hair cells and other cell types. What about the other 4? Please provide the counting data to show survival. This data should be used to correlate with hearing test results at 6 months. If cell survival protection is minimal, it should be discussed why they saw a reasonable hearing rescue at 6 months despite the lack of preservation.

ANSWER: Overall, the average ABR threshold differences between treated and untreated ears tested at six months were 18.3 ± 7.9 , 28.3 ± 10.1 , 28.3 ± 10.1 and 15.0 ± 6.2 SPL at 8, 12, 18 and 24 kHz, respectively ($p < 0.05$ for all frequencies; Fig. 6d). The differences were not statistically significant at 4 and 32 kHz ($p > 0.05$). These results suggest efficacy started to decline at 6 months. We also observed a correlation between the hearing and morphological preservations. In high-dosage group at P6m, the morphologies of the inner ears in two mice with good hearing preservation were still normal (Supplementary Fig. 8c top rows). We also observed that cellular degeneration of cochlear HCs, MCs and vestibular HCs occurred at different levels in mice with partial hearing preservation or with poor hearing preservation (Supplementary Fig. 8c, middle and bottom rows respectively).

We added the morphological data in Supplementary Fig.8c and following descriptions should directly address the Reviewer’s question:

Starting from Page9 line199: “In high-dosage group at P6m, the morphologies of the inner ears in two mice with good hearing preservation were still normal (Supplementary Fig. 8c top rows). We also observed that cellular degeneration of cochlear HCs, MCs and vestibular HCs occurred at different levels in mice with partial hearing preservation or with poor hearing preservation (Supplementary Fig. 8c, middle and bottom rows respectively). These results suggested a correlation between the hearing and morphological preservations and support that the treatment effect were retained for six months in some of the treated *Kcne1*^{-/-} mice, although the reason for variation in hearing preservation is unclear.”

Starting from Page17 line388: “In contrast, both morphologies of inner ears (Supplementary Fig. 8c) and the ABR thresholds (Fig. 6d) in two mice received the high-dosage injections were still preserved at P6m. More cellular degeneration appeared in the cochlear HCs, MCs and vestibular HCs in mice that showed partial hearing preservation. The degree of cellular degeneration appeared to be correlated with severity of hearing loss (Supplementary Fig. 8c).”

In the rebuttal, they mentioned that when *Kcne1*(-/-) pups were nursed by a WT mother, the abnormalities they observed went away. The data need to be presented.

ANSWER: Comparing to WT babies, the weight growth in *Kcne1*^{-/-} pups was normal from 4 to 8 weeks after birth when they were nursed by WT mothers. However, their hearing and vestibular functions were not improved and were similar to that we observed when mice were nursed by *Kcne1*^{-/-} mothers. We have presented the weight growth data of *Kcne1*^{-/-} pups nursed by WT mothers (4 to 8 weeks) in Supplementary Fig. 9a.

Starting from Page12 line273: “Compared to that of WT mice, the untreated *Kcne1*^{-/-} mice nursed by *Kcne1*^{-/-} mothers had a lower body weight at four (14.2 ± 0.4 g) and six (18.3 ± 0.4 g) weeks of age (n=6, p=0.007 at 4 weeks, p=0.02 at 6 weeks, student’s *t* tests). In contrast, the untreated *Kcne1*^{-/-} mice nursed by WT mothers (although their hearing and vestibular functions were not improved), and the treated *Kcne1*^{-/-} mice in both low- and high-dosage groups nursed by WT mothers, showed similar weight growth comparing to WT mice during 8-week period (n=6, p>0.05 in all comparisons, student’s *t* tests; Supplementary Fig. 9a).”

First sentence in page 18, it is overstated, as the issues discussed are only related to mice. Should change the sentence to describe the relevance in mice.

ANSWER: We have deleted the sentence.

In the discussion the potential issues need to be discussed that include 1). Identify AAV that can infect adult MC; 2). The need to address if a window of opportunity for intervention still exists in human due to rapid degeneration of relevant cells; 3) If the injection route is still available in human patients if the inner ear structure is severely damaged.

ANSWER: We have expanded the Discussion to address the Reviewer’s comments (starting from Page17 line399): “Considering that AAV1-CB7-*GFP* could not transduce the adult MCs of mice in the present study, it seems to be necessary to identify new serotypes of AAVs or other means to transduce adult MCs in future human translational studies. Our results suggested that gene therapy implemented after degeneration of cochlear and vestibular cells is unlikely to be successful if the cells in the inner ear are already degenerated, no matter which injection route is utilized. Due to the very limited availability of materials, the pathological changes about the morphologies in the cochleae of JLNS2 patients still remain unknown. We are not sure whether therapeutic efficacy can be achieved by gene therapy in the cochlea of newborn babies of JLNS2 patients. However, the postnatal window of efficacy to achieve gene therapy in the mouse model suggests that the window of therapeutic efficacy in humans may be in the second trimester (around 18 weeks gestational age in humans) and prior to hearing onset in humans. This implication may generate additional serious social and ethical issues for the human treatment given the nonlethal nature of the disease. ”

Please use * to indicate significance in the figures, as some of them do not show significant changes.

ANSWER: We have gone through the manuscript and consistently used “*” to indicate significance in statistical tests in the figures. We added “*” in Supplementary Fig. 5b-d in the 2nd revised manuscript as suggested.

Line 20 p2, “Results showed the treatment” should be “Results showed early treatment”

Line 23, p2, “(16 out of 20 mice, or 16/20) “ should be (16 out of 20 mice)”

ANSWER: We have re-written these sentences as suggested.

Reviewer #2 (Remarks to the Author):

The authors have addressed most of my comments satisfactorily. However, they have not appropriately addressed the following:

"3) Avoid use of terms such as "significant increases" (line 247). If the authors are referring to statistical significance, it is better to instead state solely "increases" and add n number, P value and effect size afterward so that readers can assess for themselves.

ANSWER: In the revised and throughout the manuscript, we only used “significant increase” when p value is provided and indicated such for the results.

This is the opposite of what I was suggesting. Many prominent statisticians are advising that the use of the term "significant" be AVOIDED COMPLETELY in this context because it is essentially meaningless. Please remove the word and instead describe the effect, e.g., X was increased xx-fold (P = xx; n = xx). Also the authors still use the term "significant" in many other parts of the manuscript where they do not quote P values, such as in the abstract and introduction. The term really is misleading.

ANSWER: We are very grateful for the reviewer's advice. We have deleted the term “significant” in the abstract, introduction and results sections and re-written these sentences as suggested.

Reviewers' Comments:

Reviewer #1:

Remarks to the Author:

Fig. 1c, the scale bar seems to be off as it'd indicate each cell surface as outlined by phalloidin is close to 25 μ m. Please double check it.

Fig. 1n, it is impossible to see co-localization of NF200/GFP. They need to show an image with NF200 and DAPI in addition to 1n.

In general the signals in the red channel are weak, making it very difficult to see clearly what's been shown. Please adjust the intensity properly for better illustration.

Line118, I can see GFP in the IHCs but not in the supporting cells. Please remove supporting cells from the parenthesis.

Line 200, "two mice with good hearing" should be "two mice with partial hearing preservation" as hearing is not good but detectable.

One additional explanation for the lack of hearing preservation at 6 mon could be due to ectopic expression of Kcne1 with toxic effect over time in cells the gene is not expressed endogenously, such as hair cells. One way to test is to use the Kcne1 promoter to drive the expression. This possibility should be discussed.

Original Reviewer 1 comments

Fig. 1c, the scale bar seems to be off as it'd indicate each cell surface as outlined by phalloidin is close to 25 μ m. Please double check it.

Fig. 1n, it is impossible to see co-localization of NF200/GFP. They need to show an image with NF200 and DAPI in addition to 1n.

In general the signals in the red channel are weak, making it very difficult to see clearly what's been shown. Please adjust the intensity properly for better illustration.

Line 118, I can see GFP in the IHCs but not in the supporting cells. Please remove supporting cells from the parenthesis.

Line 200, "two mice with good hearing" should be "two mice with partial hearing preservation" as hearing is not good but detectable.

One additional explanation for the lack of hearing preservation at 6 mon could be due to ectopic expression of Kcne1 with toxic effect over time in cells the gene is not expressed endogenously, such as hair cells. One way to test is to use the Kcne1 promoter to drive the expression. This possibility should be discussed.

Point-to-point answers to comments by Reviewer#1:

Fig. 1c, the scale bar seems to be off as it'd indicate each cell surface as outlined by phalloidin is close to 25 μ m. Please double check it.

-----Answer: As suggested, we have double checked and adjusted the scale bar in Fig 1c.

Fig. 1n, it is impossible to see co-localization of NF200/GFP. They need to show an image with NF200 and DAPI in addition to 1n.

-----Answer: To better show co-localization, we have split the original Fig 1n into two panels (new Fig 1n (red channel, labeled with the antibody against NF200) and Fig 1n' (green channel of the same image, labeled with the antibody against the GFP)). We then superimposed the two images into the new Fig 1n''. We believe the

co-localization of NF200 and GFP in SGNs is better shown in this manner (examples are pointed out by arrows in the Fig 1n”). We also rewrote the legend of the Figure 1 (Page26 line 626-631) to reflect the changes we made in the figure 1. All changed texts in this revision are given in red in the revised manuscript.

In general the signals in the red channel are weak, making it very difficult to see clearly what’s been shown. Please adjust the intensity properly for better illustration.

-----Answer: As suggested, we have adjusted and balanced the intensity of the red and green channels in the Fig1 m, n-n” to better illustrate the co-labeling.

Line118, I can see GFP in the IHCs but not in the supporting cells. Please remove supporting cells from the parenthesis.

-----Answer: As suggested, we have removed “supporting cells” from the parenthesis (line 118).

Line 200, “two mice with good hearing” should be “two mice with partial hearing preservation” as hearing is not good but detectable.

-----Answer: We have revised this sentence on line 199 (page9) to: “..... two mice with partial hearing preservation”, as suggested.

One additional explanation for the lack of hearing preservation at 6 mon could be due to ectopic expression of *Kcne1* with toxic effect over time in cells the gene is not expressed endogenously, such as hair cells. One way to test is to use the *Kcne1* promoter to drive the expression. This possibility should be discussed.

-----Answer: We have revised manuscript and add the possible explanation as suggested by the Reviewer#1 (Page17 line392-396): “Possible explanations for the reason of efficacy decline may include a gradual decline of the virally mediated *Kcne1* expression or excessive ectopic expression of *Kcne1* with toxic effect over time in cells that *Kcne1* is not expressed endogenously, such as HCs. One way to test this hypothesis may use the *Kcne1* promoter to drive the expression in the inner ear.”.